# Synthesis and Study of Dibenzo[*b, f*]oxepine Combined with Fluoroazobenzenes—New Photoswitches for Application in Biological Systems

**DOI:** 10.3390/molecules27185836

**Published:** 2022-09-08

**Authors:** Filip Borys, Piotr Tobiasz, Jakub Sobel, Hanna Krawczyk

**Affiliations:** 1Department of Organic Chemistry, Faculty of Chemistry, Warsaw University of Technology, Noakowskiego 3, 00-664 Warsaw, Poland; 2Laboratory of Cytoskeleton, Cilia Biology Nencki Institute of Experimental Biology, Polish Academy of Sciences, 3 Pasteur Street, 02-093 Warsaw, Poland

**Keywords:** dibenzo[*b, f*]oxepine derivatives, NMR and UV-VIS spectra, photopharmacology

## Abstract

Dibenzo[*b, f*]oxepine derivatives are an important scaffold in natural, medicinal chemistry, and these derivatives occur in several medicinally relevant plants. Two dibenzo[*b, f*]oxepines were selected and connected with appropriate fluorine azobenzenes. In the next step, the geometry of ***E***/***Z*** isomers was analyzed using density functional theory (DFT) calculations. Then the energies of the HOMO and LUMO orbitals were calculated for the ***E*/*Z*** isomers to determine the HOMO-LUMO gap. Next, modeling of the interaction between the obtained isomers of the compounds and the colchicine α and β-tubulin binding site was performed. The investigated isomers interact with the colchicine binding site in tubulin with a part of the dibenzo[*b, f*]oxepine or in a part of the azo switch, or both at the same time. Based on the *UV-VIS* spectra, it was found that in the case of compounds with an azo bond in the meta position, the absorption bands n→π* for both geometric isomers and their separation from π→π* are visible. These derivatives therefore have the potential to be used in photopharmacology.

## 1. Introduction

Dibenzo[*b, f*]oxepine derivatives are an important scaffold in natural, medicinal chemistry, and these derivatives occur in several medicinally relevant plants [1,2,3]. The dibenzo [*b, f*] oxepine system can be associated with various biological properties such as antidepressant and *anti*-estrogenic [4], analgesic [5], *anti*-inflammatory [6], antipsychotic [7], angiotensin II receptor antagonistic [8], antioxidant [9], antimycobacterial [10], antidiabetic [11], and antitumor activities [3,12], as well as *anti*-apoptosis [13] properties. The treatment of progressive neurodegenerative diseases [14] such as Parkinson’s and Alzheimer’s diseases [15] with synthetic dibenzo[*b, f*]oxepine derivatives is of particular interest. Dibenzo[*b, f*]oxepine derivatives, despite their various documented valuable biological properties, can cause side effects on the body [16]. It is, therefore, necessary to search for methods of reducing their negative effects on normal, healthy cells. One of them is the synthesis of hybrids of dibenzo[*b, f*]oxepine derivatives containing an azo bond in azobenzene molecules. These compounds can be used in photopharmacology as photochromic molecular switches. Photopharmacology covers the projects, synthesis, and application of drugs whose activity can be regulated with light. It is also a minimally invasive treatment method using visible light and allowing selective destruction of cancer cells [17]. A very interesting idea was presented by Heck’s team [18,19] and expanded by Feringa and Szymański [20,21,22]. The implementation of halogens (fluorine or chlorine) atoms to azo molecules makes it possible to separate the n→π* for stereoisomers ***E*** and ***Z*** absorption bands in the *VIS* part of the *UV-VIS* spectrum and to separate them from the π→π* band, which are in the *UV* part of the spectrum. This enables selective analysis of each geometric isomer and its selective activation [18,23]. In turn, red-shifted azobenzene molecules have emerged as useful molecular photoswitches to expand potential applications in photopharmacology. Fluorine linked with azobenzenes is well-compatible for the projecting of visible-light-responsive systems, ensuring stable and bidirectional photoconversions and tissue-compatible characteristics [23]. The combination of photoswitchable molecules with various materials [24] resulted in the use of green and red light to control e.g., chirality control in nanoporous materials [25], modulating thermal polymer properties [26,27], ion receptors [28], channel activity [29], peptide conformation [30], antibiotic potency [21], and the function of nucleic acids [31], tissue-compatible characteristics [23], and for controlling the transport through biological membranes [32]. Continuing our investigations with hybrid molecules [33], the connection of these two molecules into one system (dibenzo[*b, f*]oxepine and azobenzenes) is the subject of the presented investigations for better interaction on microtubules as anticancer compounds. Why action on microtubules? Microtubules (MTs) highly dynamic structures composed of α- and β-tubulin heterodimers, are involved in cell movement and intracellular traffic and are essential for cell division, thus, the MT skeleton is an important target for anticancer therapy. Ligands that target microtubules and affect their dynamics belong to the most successful classes of chemotherapeutic drugs against cancer by inhibiting cell proliferation. The tubulin heterodimer contains at least six distinct drug-binding sites: taxane, vinca, maytansine, and laulimalide/peloruside sites located on β-tubulin and the colchicine site located near the intradimer interface between the α- and β-tubulin subunits, while the pironetin site is a binding pocket located on the α-tubulin surface. For the first three of these sites, drugs are in current use in clinical oncology [34]. Colchicine itself binds to tubulin very tightly, but its severe toxicity to normal tissues has hindered its use in the clinic. Over the last decades, a large number of compounds able to interact with the colchicine binding site have been reported [35,36]. Derivatives of combretastatin: A-4 (CA-4), and A1(CA-1) are some of the most promising *anti*-tubulin agents that target the colchicine site [37,38,39].

Photochromic *anti*-tubulin agents have dynamically emerged in the last decade. An example of this is photoswitchable isosteric analogs of combretastatin A-4 (azo- cobretastatins- reversibly photoswitchable analogs of combretastatin A-4, named photostatins) [40]. Photostatins switch microtubule dynamics off and on under blue and green light and control mitosis in vivo with spatial precision on the single-cell level. These issues have been intensively studied by Thorn-Seshold’s team, [41,42,43] which analyzed the structural and steric interactions between different molecules (the hemithioindigo or styrylbenzothiazoles) and the binding site in tubulin. In 2020 [44], they developed photoswitchable paclitaxel (Paclitaxel, trade name Taxol, is a chemotherapy medication used to treat several types of cancer. This includes ovarian, breast, lung, Kaposi’s sarcoma, cervical, and pancreatic cancer) [45]. The photoisomerization of this reagent in living cells has been shown to allow optical control over microtubule network integrity and dynamics, cell division, and survival, with the biological response on the timescale of seconds and spatial precision to the level of individual cells within a population. Interesting research was presented by Arndt et al. [46] with a family of azobenzene-based small molecules, termed optojasps, that provide direct optical control of the actin cytoskeleton. For an overview of the research with photochromic antimitotic agents, refer to the recently published review article by Kirchner and Pianowski [47].

This manuscript is a continuation of our investigations with hybrid molecules interacting with microtubules [48]. Therefore, the goal of our research was to check the potential interaction of dibenzo[*b, f*]oxepine derivatives on tubulin (with the colchicine binding site) linked to azo molecules allowing the separation of n→π* absorption bands of ***E*** and ***Z*** stereoisomers in the *VIS* part *UV-VIS* spectrum and their separation from the π→π* band, which are in the part of the *UV* spectrum.

## 2. Results and Discussion 

To study the activity of the photoswitchable compounds of dibenzo[*b, f*]oxepine, we introduced fluoro substituted diphenyldiazene that may be in ***E*** or ***Z*** configuration. Before the synthesis of azodibenzo[*b, f*]oxepine derivatives, we performed a computational study (see Appendix A) because dibenzo[*b, f*]oxepine also has a similar spatial structure to colchicine and combretastatine (Figure 1). 

**Computational aspects.** We analyzed the geometry of the ***E****/**Z*** isomers (**4a–4h**) and (**5a–5h**) in the azo molecules (Figure 1) because the comparison between the ***E***- and ***Z***-configuration shows the energetic impact of the isomerization.

For this purpose, calculations of the density functional theory (DFT) were used (Table 1). In the calculations, the B3LYP functional and 6-31G* basis set was employed and the continuum model (PCM; Gaussian 03W, see Appendix A) was used to simulate the effects of the solvent, DMSO. The SCF energy for the ***E*** isomers was in the range of 64.71–44.13 kJ/mol, which was greater than the range for the ***Z*** isomers. We can observe in *para* and *meta* substituted azo compounds (**4b**,**4c**,**4f**,**4g**) (**5a–5h**) that the substituent in dibenzo[*b, f*]oxepine does not affect the difference energy between corresponding ***E*/*Z*** isomers: 57.65 kJ/mol for (**4b**) and 58.40 kJ/mol for (**4f**); 47.50 kJ/mol for (**4c**) and 47.68 kJ/mol for (**4g**); 56.21 kJ/mol for (**5a**) and 56.08 kJ/mol for (**5e**); 58.49 kJ/mol for (**5b**) and 58.39 kJ/mol for (**5f**); 62.09 kJ/mol for (**5c**) and 64.71 kJ/mol for (**5g**); 44.58 kJ/mol for (**5d**) and 47.06 kJ/mol for (**5h**). The Δ total energy for the same substituent in the azo switches is close. The deviation can be observed in pairs (**4a/4e**) and (**4d/4h**), resulting from the different settings of the methoxy group. Based on the quantum mechanical calculations, it can be concluded that the ***E*** and ***Z*** isomers of obtained compounds, containing fluorine atoms in *ortho*-position to the azo bond, are characterized by the lowest values of internal energy, which is associated with an easier transition between two isomeric forms. 

In the next step, we calculated the energies of the HOMO and LUMO orbitals for the ***E***/***Z*** isomers (**4a–4h**) and (**5a–5h**) (Table 2). Our approach to the design of active switches in visible light is to extend the coupling system of a known compound to lower the HOMO-LUMO gap and thus shift the excitation towards red radiation. Excitation to an active form of molecule (often ***Z*** form) in harmless visible light is one of the goals of photopharmacology [17]. Direct photoexcitation of molecules in which an appropriate gap for the HOMO-LUMO orbitals is arranged (e.g., by inserting different substituents in our case fluorine) leads to activation by visible light. It is known that *ortho*-fluorine atoms reduce the electron density in the nearby N=N bond, hence lowering the n-orbital energy [19]. In addition, fluorine atoms have a relatively small radius, and they hardly distort the planar geometry of *E*-azobenzene (dihedral angle for (**4a–4h**), and (**5a–5h**) ***E*** isomers are in the range of 180° to 179.7° (Appendix A). Electron withdrawing groups (EWGs) have been incorporated in *para* or *meta* positions (peptide bonds with dibenzo[*b, f*]oxepines) to obtain the spectral features, i.e., maxima of absorption and separation of the n→π* bands. The HOMO-LUMO gaps of (**4a–4h**) and (**5a–5h**) are in the ranges of 3.851–3.663 eV and 4.087–3.78 eV, respectively, and are similar to the gap for azobenzene (3.95 eV for *E* and 3.77 eV for ***Z*** isomer) [19]. This strategy of addition EWGs, guided by molecular orbital (MO) theory calculations, allows for photoswitching in both directions via n→π* excitation in the visible range of the spectrum. 

Molecular docking. Next, we modeled the interaction between the isomers, (**4a–4h**) and (**5a–5h**), and the colchicine binding site of α and β-tubulin (Appendix A and Table 3). The molecular docking of the compounds of the (**4a–4h**) and (**5a–5h**) isomers, ***E*** and ***Z***, into the 3D X-ray structure of tubulin (PDB code: 1SA0) [49] was carried out using the AutoDock Vina software (the Broyden–Fletcher–Goldfarb–Shanno (BFGS) method) [50]. The configurations of the protein/hybrid dimethoxydibenzo[*b, f*]oxepine combined with fluoroazobenzenes complex were created using UCSF Chimera software [51]. The graphical user interface, ADT, was employed to set up the enzyme, and all the hydrogens were added. For macromolecules, the generated pdbqt files were saved. The 3D structures of ligand molecules were built, optimized (a B3LYP functional and 6–31* basis set level) for the(**4a–4h**) and (**5a–5h**) ***E***, ***Z*** isomers, and saved in Mol2 format. The graphical user interface, ADT, was also employed to set up the ligand, and the pdbqt file was saved. The AutoDock Vina software was employed for all docking calculations. The AutoDockTools program was used to generate the docking input files. During docking, a grid box of size 21 × 21 × 21 points in the x, y, and z directions was built, and the maps were centrally located (39.82, 53.24, −8.21) in the catalytic site of the protein. A grid spacing of 0.375 Å (approximately one–fourth of the length of a carbon-carbon covalent bond) was used for the calculation of the energetic map.

The structures of (**4a–4h)** and (**5a–5h**) isomers, ***E*** and ***Z***, had an estimated binding energy presented in Appendix A (the binding energy of the colchicine is −36.0 kJ/mol/−8.6 kcal/mol) [52,53]. The model was similar to the models between colchicine, and the colchicine binding site in tubulin [54,55]. It is worth noting that for all the compounds obtained, the binding energy in the complex with tubulin was lower than for the colchicine itself. Furthermore, isomers ***Z***: (**4c*Z***–**4f*Z***)*,* (***4*h*Z***) and (**5b*Z***), (**5c*Z***), (**5e*Z***–**5h*Z***) predominantly form complexes with the dibenzo[*b, f*]oxepine part, whereas isomers ***E***: (**4a*E***–**4f*E***) and (**5a*E***–**5g*E***) act by the azo switch part (see Appendix A). It is a consequence of the spatial structure of the hybrid. As the system is in the ***Z*** configuration, the azo part of the switch does not connect to the colchicine binding site. This is well visible for compounds where the azo bond is in the *meta* position; then all ***Z*** isomers form a complex with tubulin through the dibenzo[*b, f*]oxepine part. Exceptions appear for (**4a*Z***), (**4b*Z***), and (**4g*Z***), as these compounds form a complex with colchicine binding site in tubulin through the azo part too, and for (**5a*Z***), (**5d*Z***) in which part of the dibezo[*b, f*]oxepine and part of the azobenzene both interact with the colchicine binding site (see Appendix A). However, the isomers of (**4g*E***) and (**4h*E***) form complexes with the dibenzo[*b, f*]oxepine part. We can conclude that part of the dibenzo [*b, f*]oxepine, part of the azo switch, and even both of them together, interact with the colchicine binding site in tubulin. The differences result from the spatial arrangement of the molecule. In the binding pose, compounds (**4c*Z****)*, (**4d*E****)*, (**4h*E****)*, (**5b*E****)*, and (**5f*E****)* interact with the colchicine binding site via hydrophobic interaction and hydrogen bonding; moreover, binding is stabilized by a halogen bond or π – interactions. Based on the docking results residues, Glyn11, Cys241, Asn101, and Thr202 are responsible for halogen bond interactions. Active residues (non-hydrophobic interactions) are shown in Table 3 (for detailed interactions for all ligands, see Appendix A).

### 2.1. Synthesis

Based on the data from the calculations, we would expect that azodibenzo[*b, f*]oxepine derivatives would be a potent tubulin inhibitor; therefore, we synthesized and investigated a set of the azo compounds (Table 1 and Figure 1). The purpose of this work was to develop a method for the synthesis of dibenzo[*b, f*]oxepine derivative hybrids containing an azo bond with the potential to be photochromic molecular switches in photopharmacological therapies. The azo compounds with a carboxyl group were transformed into more reactive acid chlorides with thionyl chloride. The amides—hybrids containing the dibenzo[*b, f*]oxepine backbone with an azo bond—were synthesized from the amine derivatives of dibenzo[*b, f*]oxepine and the obtained acid chlorides.

### 2.2. NMR and UV-VIS Spectra

In the NMR spectra of the products (**4a–4h**) and (**5a–5h**), we observed two sets of signals with very similar chemical shifts (see Appendix A). First, we tried to prove that the products are not atropoisomers- *syn* or *anti*-form but***E*** and ***Z*** isomers. Therefore, we measured the spectra for substrates (**1a–1d**) and (**2a–2d**) and observed one set of signals for each compound. Next, we made the ^1^H NMR spectra for (**5h**) products at different temperatures (80 °C–150 °C, Appendix A). The interconversion of compounds can be monitored by variable-temperature NMR spectroscopy (dynamic NMR) when the reaction is slow on the NMR time scale. While raising the temperature of the DMSO solution of the (**5h**) products, we expected the proton signals to broaden and coalesce, eventually yielding a single set of lines for products (**5h**), owing to the rapid interconversion of the two atropoisomers at high temperatures, but this did not occur (Appendix A). During the irradiation of (**5h**) in an NMR tube with the light at a wavelength of λ = 525 nm/ 1 h, it obtained one set of signals, the same as in 150 °C (***E***-to-***Z*** photoisomerization, the ratio ***E*/*Z***, see Appendix A). These experiments confirm the presence of ***E*** and***Z*** isomers in all products at room temperature (the ***E*** isomer prevails in solution, the ratio ***E*/*Z***, see Appendix A). From the calculations, it can be observed that the conformations for all (**4a–4h**) and (**5a–5h**) ***E*** and ***Z*** isomers between the dibenzo[*b, f*]oxepine and azobenzene rings are the same (Figure 2, Table 1). This is due to the formation of a hydrogen bond between the oxygen of the carbonyl group and one of the *ortho*- protons in the dibenzo [*b, f*] oxepine ring. The calculated distances between these atoms range from 2.243 to 2.209 Å. Moreover, in this conformation, the largest substituents are located furthest from each other: azobenzene and the dibenzo[*b, f*]oxepine substituent, and therefore, *anti-*form probably prevails in structure (Figure 2). Moreover, we performed calculations for the (**5h)** compound in the *syn* form (we froze the NH-CO bond). The difference in energy between *anti* and *syn* was 0,00705851 au (18.63 kJ/mol, Table 1) and is smaller than between ***E*** and ***Z*** isomers. In gated ^13^C NMR (irradiation of one of the protons *ortho* of dibenzo [*b, f*] oxepine (**4h*E***) in the ring closer to the carbonyl group), we observed dipolar interaction between proton *ortho* and carbon of carbonyl group (NOE effect of about 9%). This experiment confirmed that *anti*-form dominates in the (**4h*E*/4hZ**) structure and we can postulate for the remaining products that this form dominates in the solution.

**4a.** yield 46%; time of reaction 0.5 h; orange powder mp = 215.5 °C;

Isomer *E:*
^1^H NMR (500 MHz, DMSO-*d*_6_, 298 K): δ (ppm): Dibenzo[*b, f*]oxepine: 10.57 (1H, s, NH), 7.79 (1H, d, *J*_H2,H4_ = 1.5 Hz, H4), 7.66 (1H, dd, *J*_H1,H2_ = 8.5 Hz, H2), 7.27 (1H, d, H1), 7.12–7.08 (2H, m, H7, H8), 6.83 (1H, dd, *J*_H8,H9_ = 6.5 Hz, *J*_H7,H9_ = 3 Hz, H9), 6.78 (1H, AB spin system, d, *J*_H10,H11_ = 11.5 Hz, H10), 6.72 (1H, AB spin system, d, H11), 3.87 (3H, s, OCH_3_); Azo: 8.18 (2H, d, *J*_H2,H3_ = 8.5 Hz, H3), 8.02 (2H, d, H2), 7.95 (2H, dd, *J*_H2’,H3’_ = 8 Hz, *J*_H2’,H4’_ = 2 Hz, H2’), 7.65–7.61 (3H, m, H3’, H4’).

Isomer *Z:*
^1^H NMR(500 MHz, DMSO-*d*_6_, 298 K): δ (ppm): Dibenzo[*b, f*]oxepine: 10.37 (1H, s, NH), 7.71 (1H, d, *J*_H2,H4_ = 1.5 Hz, H4), 7.57 (1H, dd, *J*_H1,H2_ = 8.5 Hz, H2), 6.98 (1H, d, H1), 6.90–6.88 (2H, m, H7, H8), 6.81 (1H, dd, *J*_H8,H9_ = 6.5 Hz, *J*_H7,H9_ = 3 Hz, H9), 6.76 (1H, AB spin system, d, *J*_H10,H11_ = 11.5 Hz, H10), 6.70 (1H, AB spin system, d, H11), 3.85 (3H, s, OCH_3_); Azo: 8.18 (2H, d, *J*_H2,H3_ = 8.5 Hz, H3), 7.88 (2H, d, H2), 7.83 (2H, dd, *J*_H2’,H3’_ = 8 Hz, *J*_H2’,H4’_ = 2 Hz, H2’), 7.65–7.61 (3H, m, H3’, H4’).

Ratio *E*/*Z* 20:1

Isomer *E**:* ^13^C NMR (125 MHz, DMSO-*d*_6_, 298 K): δ (ppm): 164.75, 164.53, 156.62, 156.56, 156.26, 153.53, 153.49, 151.92, 151.48, 151.45, 144.20, 144.17, 140.85, 136.74, 132.71, 132.11, 131.48, 131.47, 129.68, 129.57, 129.25, 129.17, 129.14, 128.92, 128.77, 128.69, 128.61, 127.59, 126.09, 125.95, 125.09, 125.06, 122.79, 122.40, 120.54, 120.51, 120.02, 119.52, 116.63, 116.53, 113.25, 113.16, 113.00, 112.97, 56.16, 56.13.

HRMS (ESI): *m*/*z* calculated for C_28_H_21_N_3_O_3_, 447.15774; found: 447.15776.

**4b**. yield 76%; time of reaction 0.5 h; orange powder mp = 242.5 °C; 

Isomer *E*: ^1^H NMR (500 MHz, DMSO-*d*_6_, 298 K): δ (ppm): Dibenzo[*b, f*]oxepine: 10.57 (1H, s, NH), 7.78 (1H, d, *J*_H2,H4_ = 2.5 Hz, H4), 7.66 (1H, dd, *J*_H1,H2_ = 8.5 Hz, H2), 7.27 (1H, d, H1), 7.10–7.08 (2H, m, H7, H8), 6.83 (1H, dd, *J*_H8,H9_ = 6.5 Hz, *J*_H7,H9_ = 2.5 Hz, H9), 6.78 (1H, AB spin system, d, *J*_H10,H11_ = 11.5 Hz, H10), 6.72 (1H, AB spin system, d, H11), 3.87 (3H, s, OCH_3_); Azo: 8.18 (2H, d, *J*_H2,H3_ = 8.5 Hz, H2), 8.03 (2H, dd, *J*_H2’,H3’_ = 8.5 Hz, *J*_H2’,F_ = 5 Hz, H2’), 8.01 (2H, d, H2), 7.47 (2H, t, *J*_H3’,F_ = 8.5 Hz, H3’).

Isomer *Z*: ^1^H NMR (500 MHz, DMSO-*d*_6_, 298 K): δ (ppm): Dibenzo[*b, f*]oxepine: 10.38 (1H, s, NH), 7.71 (1H, d, *J*_H2,H4_ = 2.5 Hz, H4), 7.58 (1H, dd, *J*_H1,H2_ = 8.5 Hz, H2), 7.23 (1H, d, H1), 7.00–6.96 (2H, m, H7, H8), 6.82 (1H, dd, *J*_H8,H9_ = 6.5 Hz, *J*_H7,H9_ = 2.5 Hz, H9), 6.75 (1H, AB spin system, d, *J*_H10,H11_ = 11.5 Hz, H10), 6.70 (1H, AB spin system, d, H11), 3.85 (3H, s, OCH_3_); Azo: 8.18 (2H, d, *J*_H2,H3_ = 8.5 Hz, H2), 8.03 (2H, dd, *J*_H2’,H3’_ = 8.5 Hz, *J*_H2’,F_ = 5 Hz, H2’), 7.90 (2H, d, H2), 7.18 (2H, t, *J*_H3’,F_ = 8.5 Hz, H3’). 

Ratio *E*/*Z* 2.5:1 

Isomer *E*: ^13^C NMR (125 MHz, DMSO-*d*_6_, 298 K): δ (ppm): 165.15, 164.73, 163.15, 156.61, 156.09, 153.40, 151.47, 148.72, 148.70, 144.20, 140.84, 136.76, 131.48, 129.68, 129.25, 129.14, 128.77, 128.74, 128.70, 126.09, 125.26, 125.19, 125.09, 122.75, 122.68, 122.39, 120.54, 119.43, 116.68, 116.62, 116.54, 116.50, 115.95, 115.77, 113.24, 113.17, 113.00, 56.15.

HRMS (ESI): *m*/*z* calculated for C_28_H_20_FN_3_O_3_, 465.14832; found: 465.14843

**4c**. yield 26%; time of reaction 0.5 h; orange powder mp = 246 °C; 

Isomer *Z*: ^1^H NMR (500 MHz, DMSO-*d*_6_, 298 K): δ (ppm): Dibenzo[*b, f*]oxepine: 10.58 (1H, s, NH), 7.78 (1H, d, *J*_H2,H4_ = 2 Hz, H4), 7.66 (1H, dd, *J*_H1,H2_ = 8.5 Hz, H2), 7.27 (1H, d, H1), 7.09 (1H, t, *J*_H8,H9_,7 = 6.5 Hz, H8), 7.08 (1H, dd, *J*_H7,H9_ = 2.5 Hz, H7), 6.83 (1H, dd, H9), 6.78 (1H, AB spin system, d, *J*_H10,H11_ = 11.5 Hz, H10), 6.72 (1H, AB spin system, d, H11), 3.87 (3H, s, OCH_3_); Azo: 8.18 (2H, d, *J*_H2,H3_ = 8.5 Hz, H3), 8.01 (2H, d, H2), 7.87 (1H, td, *J*_H5’,H6’_ = 9 Hz, *J*_H5’,F_ = 6.5 Hz H6’), 7.62 (1H, ddd, *J*_H3’,F_ = 9 Hz, *J*_H3’,H5’_ = 3 Hz, H3’), 7.32–7.30 (1H, m, H5’).

Isomer *E*: ^1^H NMR (500 MHz, DMSO-*d*_6_, 298 K): δ (ppm): Dibenzo[*b, f*]oxepine: 10.42 (1H, s, NH), 7.71 (1H, d, *J*_H2,H4_ = 2 Hz, H4), 7.58 (1H, dd, *J*_H1,H2_ = 8.5 Hz, H2), 7.23 (1H, d, H1), 7.09 (1H, t, *J*_H8,H9_, 7 = 6.5 Hz, H8), 7.08 (1H, dd, *J*_H7,H9_ = 2.5 Hz, H7), 6.82 (1H, dd, H9), 6.75 (1H, AB spin system, d, *J*_H10,H11_ = 11.5 Hz, H10), 6.70 (1H, AB spin system, d, H11), 3.85 (3H, s, OCH_3_); Azo: 8.18 (2H, d, *J*_H2,H3_ = 8.5 Hz, H3), 7.92 (2H, d, H2), 7.87 (1H, td, *J*_H5’,H6’_ = 9 Hz, *J*_H5’,F_ = 6.5 Hz, H6’), 7.62 (1H, ddd, *J*_H3’,F_ = 9 Hz, *J*_H3’,H5’_ = 3 Hz, H3’), 7.18–7.13 (1H, m, H5’). 

Ratio *E*/*Z* 2.5:1 

Isomer *E*: 13C NMR (125 MHz, DMSO-d6, 298K): δ (ppm): 164.69, 164.45, 156.61, 156.56, 156.10, 153.54, 151.47, 151.45, 144.19, 144.17, 140.81, 140.79, 137.16, 133.73, 131.48, 131.47, 129.68, 129.66, 129.25, 129.21, 129.19, 128.78, 128.72, 126.12, 126.01, 125.09, 125.07, 122.61, 120.54, 120.52, 119.34, 119.25, 119.09, 116.64, 116.55, 113.25, 113.18, 113.00, 112.98, 112.79, 112.60, 105.97, 105.76, 105.57, 56.15, 56.13.

HRMS (ESI): *m*/*z* calculated for C_28_H_19_F_2_N_3_O_3_, 483.13890; found: 483.13884

**4d**. yield 16%; time of reaction 0.5 h; orange powder mp = 230 °C; 

Isomer *E*: ^1^H NMR (500 MHz, DMSO-*d*_6_, 298 K): δ (ppm): Dibenzo[*b, f*]oxepine: 10.59 (1H, s, NH), 7.78 (1H, d, *J*_H2,H4_ = 2 Hz, H4), 7.65 (1H, dd, *J*_H1,H2_ = 8.5 Hz, H2), 7.27 (1H, d, H1), 7.08 (1H, t, *J*_H8,H7,9_ = 6 Hz, H8), 7.07 (1H, dd, *J*_H7,H9_ = 3 Hz, H7), 6.82 (1H, dd, H9), 6.77 (1H, AB spin system, d, *J*_H10,H11_ = 11.5 Hz, H10), 6.71 (1H, AB spin system, d, H11), 3.86 (3H, s, OCH_3_); Azo: 8.19 (2H, d, *J*_H2,H3_ = 8.5 Hz, H3), 7.99 (2H, d, H2), 7.61 (1H, tt, *J*_H3’,H4’_ = 8.5 Hz, *J*_H4’,F_ = 6 Hz, H4’), 7.36 (2H, t, *J*_H3’,F_ = 9.5 Hz, H3’).

Isomer *Z*: ^1^H NMR (500 MHz, DMSO-*d*_6_, 298 K): δ (ppm): Dibenzo[*b, f*]oxepine: 10.44 (1H, s, NH), 7.70 (1H, d, *J*_H2,H4_ = 2.5 Hz, H4), 7.57 (1H, dd, *J*_H1,H2_ = 8.5 Hz, H2), 7.22 (1H, d, H1), 7.08 (1H, t, *J*_H8,H7,9_ = 6 Hz, H8), 7.07 (1H, dd, *J*_H7,H9_ = 3 Hz, H7), 6.80 (1H, dd, H9), 6.74 (1H, AB spin system, d, *J*_H10,H11_ = 11.5 Hz, H10), 6.69 (1H, AB spin system, d, H11), 3.84 (3H, s, OCH_3_); Azo: 8.19 (2H, d, *J*_H2,H3_ = 8.5 Hz, H3), 7.93 (2H, d, H2), 7.61 (1H, tt, *J*_H3’,H4’_ = 8.5 Hz, *J*_H4’,F_ = 6 Hz, H4’), 7.12 (2H, t, *J*_H3’,F_ = 9.5 Hz, H3’). 

Ratio *E*/*Z* 1.5:1 

Isomer *E*: ^13^C NMR (125 MHz, DMSO-*d*_6_, 298 K): δ (ppm): 164.62, 164.39, 156.62, 156.56, 156.44, 155.96, 154.00, 153.90, 151.48, 151.45, 144.20, 144.17, 140.78, 140.76, 137.59, 134.61, 132.59, 131.48, 131.47, 130.23, 129.68, 129.66, 129.26, 129.24, 129.19, 128.87, 128.80, 128.74, 126.16, 126.05, 125.10, 125.07, 122.41, 120.54, 120.52, 118.25, 116.66, 116.56, 113.27, 113.25, 113.19, 113.09, 113.01, 112.98, 112.55, 112.37, 56.16, 56.13.

HRMS (ESI): *m*/*z* calculated for C_28_H_19_F_2_N_3_O_3_, 483.13890; found: 483.13901.

**4e**. yield 30%; time of reaction 0.5 h; orange powder mp = 168 °C; 

Isomer *E*: ^1^H NMR (500 MHz, DMSO-*d*_6_, 298 K): δ (ppm): Dibenzo[*b, f*]oxepine: 10.55 (s, 1H, NH), 7.79 (1H, d, *J*_H2,H4_ = 2 Hz, H4), 7.56 (1H, dd, *J*_H1,H2_ = 8.5 Hz, H2), 7.23 (1H, d, H1), 7.18 (1H, d, *J*_H8,H9_ = 8.5 Hz, H9), 6.80 (1H, d, *J*_H6,H8_ = 2.5 Hz, H6), 6.77 (1H, dd, H8), 6.64 (1H, AB spin system, d, *J*_H10,H11_ = 11.5 Hz, H10), 6.59 (1H, AB spin system, d, H11), 3.78 (3H, s, OCH_3_); Azo: 8.18 (2H, d, *J*_H2_,_H2_ = 8.5 Hz, H2), 8.02 (2H, d, H3), 7.95 (2H, dd, *J*_H2’,H3’_ = 8.5 Hz, *J*_H2’,H4’_ = 2 Hz, H2’), 7.63–7.62 (3H, m, H3’, H4’).

Isomer *Z*: ^1^H NMR (500 MHz, DMSO-*d*_6_, 298 K): δ (ppm): Dibenzo[*b, f*]oxepine: 10.35 (s, 1H, NH), 7.72 (1H, d, *J*_H2,H4_ = 2 Hz, H4), 7.48 (1H, dd, *J*_H1,H2_ = 8.5 Hz, H2), 6.99 (1H, d, H1), 6.89 (1H, d, *J*_H8,H9_ = 8.5 Hz, H9), 6.76 (1H, d, *J*_H6,H8_ = 2.5 Hz, H6), 6.75 (1H, dd, H8), 6.62 (1H, AB spin system, d, *J*_H10,H11_ = 11.5 Hz, H10), 6.56 (1H, AB spin system, d, H11), 3.77 (3H, s, OCH_3_); Azo: 8.18 (2H, d, *J*_H2,H2_ = 8.5 Hz, H2), 7.87 (2H, d, H3), 7.95 (2H, dd, *J*_H2’,H3’_ = 8.5 Hz, *J*_H2’,H4’_ = 2 Hz, H2’), 7.33–7.30 (3H, m, H3’, H4’). 

Ratio *E*/*Z* 3.2:1 

Isomer *E*: ^13^C NMR (125 MHz, DMSO-*d*_6_, 298 K): δ (ppm): 164.69, 161.18, 157.31, 156.07, 153.55, 151.92, 140.62, 136.71, 132.11, 130.20, 129.57, 129.30, 129.08, 128.55, 127.20, 126.02, 122.92, 122.79, 122.43, 116.84, 112.97, 111.22, 106.68, 55.52.

HRMS (ESI): *m*/*z* calculated for C_28_H_21_N_3_O_3_, 447.15774; found: 447.15764.

**4f.** yield 34%; time of reaction 0.5 h; orange powder mp = 218 °C; 

Isomer *E*: ^1^H NMR (500 MHz, DMSO-*d*_6_, 298 K): δ (ppm): Dibenzo[*b, f*]oxepine: 10.55 (1H, s, NH), 7.79 (1H, d, *J*_H2,H4_ = 2 Hz, H4), 7.56 (1H, dd, *J*_H1,H2_ = 8.5 Hz, H2), 7.23 (1H, d, H1), 7.18 (1H, d, *J*_H8,H9_ = 8.5 Hz, H9), 6.80 (1H, d, *J*_H6,H8_ = 2.5 Hz, H6), 6.77 (1H, dd, H8), 6.64 (1H, AB spin system, d, *J*_H10,H11_ = 11.5 Hz, H10), 6.59 (1H, AB spin system, d, H11), 3.78 (3H, s, OCH_3_); Azo: 8.17 (2H, d, *J*_H2,H3_ = 8.5 Hz, H3), 8.02 (2H, dd, *J*_H2’,H3’_ = 9 Hz, *J*_H2’,F_ = 5 Hz, H2’), 8.01 (2H, d, H2), 7.47 (2H, t, *J*_H3’,F_ = 9 Hz, H3’).

Isomer *Z*: ^1^H NMR (500 MHz, DMSO-*d*_6_, 298 K): δ (ppm): Dibenzo[*b, f*]oxepine: 10.36 (1H, s, NH), 7.72 (1H, d, *J*_H2,H4_ = 2 Hz, H4), 7.49 (1H, dd, *J*_H1,H2_ = 8.5 Hz, H2), 7.00 (1H, d, H1), 6.97 (1H, d, *J*_H8,H9_ = 8.5 Hz, H9), 6.77 (1H, d, *J*_H6,H8_ = 2.5 Hz, H6), 6.75 (1H, dd, H8), 6.62 (1H, AB spin system, d, *J*_H10,H11_ = 11.5 Hz, H10), 6.57 (1H, AB spin system, d, H11), 3.77 (3H, s, OCH3); Azo: 8.17 (2H, d, *J*_H2,H3_ = 8.5 Hz, H3), 8.02 (2H, dd, *J*_H2’,H3’_ = 9 Hz, *J*_H2’,F_ = 5 Hz, H2’), 7.90 (2H, d, H2), 7.18 (2H, t, *J*_H3’,F_ = 9 Hz, H3’). 

Ratio *E*/*Z* 3.5:1 

Isomer *E*: ^13^C NMR (125 MHz, DMSO-*d*_6_, 298 K): δ (ppm): 165.16, 164.67, 164.44, 163.16, 161.18, 161.15, 157.31, 157.28, 156.11, 156.07, 156.02, 153.42, 149.66, 149.64, 148.72, 148.69, 140.61, 140.50, 136.73, 136.68, 132.77, 130.20, 130.18, 129.30, 129.23, 129.08, 128.68, 128.55, 128.47, 127.20, 126.03, 125.89, 125.27, 125.20, 122.92, 122.75, 122.68, 122.43, 119.47, 116.84, 116.74, 116.69, 116.50, 115.95, 115.77, 112.97, 112.87, 111.21, 111.18, 106.68, 106.66, 55.52, 55.50.

HRMS (ESI): *m*/*z* calculated for C_28_H_20_FN_3_O_3_, 465.14832; found: 465.14824.

**4g**. yield 23%; time of reaction 0.5 h; orange powder mp = 215 °C; 

Isomer *E*: ^1^H NMR (500 MHz, DMSO-*d*_6_, 298 K): δ (ppm): Dibenzo[*b, f*]oxepine: 10.57 (1H, s, NH), 7.79 (1H, d, *J*_H2,H4_ = 2 Hz, H4), 7.56 (1H, dd, *J*_H1,H2_ = 8 Hz, H2), 7.23 (1H, d, H1), 7.18 (1H, d, *J*_H8,H9_ = 8.5 Hz, H9), 6.80 (1H, d, *J*_H6,H8_ = 2.5 Hz, H6), 6.77 (1H, dd, H8), 6.64 (1H, AB spin system, d, *J*_H10,H11_ = 11.5 Hz, H10), 6.59 (1H, AB spin system, d, H11), 3.78 (3H, s, OCH_3_); Azo: 8.18 (2H, d, *J*_H2,H3_ = 8.5 Hz, H3), 8.01 (2H, d, H2), 7.87 (1H, td, *J*_H5’,H6’_ = 8.5 Hz, *J*_H6’,F_ = 6.5 Hz, H6’), 7.62 (1H, ddd, *J*_H3’,F_ = 9 Hz, *J*_H3’,H5’_ = 2.5 Hz, H3’), 7.31–7.27 (1H, m, H5’).

Isomer *Z*: ^1^H NMR (500 MHz, DMSO-*d*_6_, 298 K): δ (ppm): Dibenzo[*b, f*]oxepine: 10.40 (1H, s, NH), 7.72 (1H, d, *J*_H2,H4_ = 2 Hz, H4), 7.48 (1H, dd, *J*_H1,H2_ = 8 Hz, H2), 7.19 (1H, d, H1), 7.05 (1H, d, *J*_H8,H9_ = 8.5 Hz, H9), 6.76 (1H, d, *J*_H6,H8_ = 2.5 Hz, H6), 6.75 (1H, dd, H8), 6.62 (1H, AB spin system, d, *J*_H10,H11_ = 11.5 Hz, H10), 6.57 (1H, AB spin system, d, H11), 3.77 (3H, s, OCH_3_); Azo: 8.18 (2H, d, *J*_H2,H3_ = 8.5 Hz, H3), 7.92 (2H, d, H2), 7.87 (1H, td, *J*_H5’,H6’_ = 8.5 Hz, *J*_H6’,F_ = 6.5 Hz, H6’), 7.62 (1H, ddd, *J*_H3’,F_ = 9 Hz, *J*_H3’,H5’_ = 2.5 Hz, H3’), 7.13–7.09 (1H, m, H5’). 

Ratio *E*/*Z* 5.5:1 

Isomer *E*: ^13^C NMR (125 MHz, DMSO-*d*_6_, 298 K): δ (ppm): 164.63, 161.19, 157.30, 157.28, 156.06, 153.58, 153.55, 140.58, 137.13, 130.20, 129.31, 129.24, 129.15, 128.67, 128.56, 127.20, 126.06, 125.94, 122.92, 122.65, 119.33, 119.25, 119.13, 116.85, 116.76, 112.99, 112.82, 112.63, 111.22, 106.68, 106.66, 105.98, 105.79, 105.57, 55.52, 55.51.

HRMS (ESI): *m*/*z* calculated for C_28_H_19_F_2_N_3_O_3_, 483.13890; found: 483.13864.

**4h**. yield 39%; time of reaction 0.5 h; orange powder mp =175 °C; 

Isomer *E*: ^1^H NMR (500 MHz, DMSO-*d*_6_, 298 K): δ (ppm): Dibenzo[*b, f*]oxepine: 10.58 (1H, s, NH), 7.79 (1H, d, *J*_H2,H4_ = 2 Hz, H4), 7.56 (1H, dd, *J*_H1,H2_ = 8.5 Hz, H2), 7.23 (1H, d, H1), 7.18 (1H, d, *J*_H8,H9_ = 8.5 Hz, H9), 6.80 (1H, d, *J*_H6,H8_ = 2.5 Hz, H6), 6.77 (1H, dd, H8), 6.64 (1H, AB spin system, d, *J*_H10,H11_ = 11.5 Hz, H10), 6.59 (1H, AB spin system, d, H11), 3.78 (3H, s, OCH_3_); Azo: 8.19 (2H, d, *J*_H2,H3_ = 8.5 Hz, H3), 8.00 (2H, d, H2), 7.62 (1H, tt, *J*_H3’,H4’_ = 8.5 Hz, *J*_H4’,F_ = 6 Hz, H4’), 7.37 (2H, t, *J*_H3’,F_ = 9 Hz, H3’).

Isomer *Z*: ^1^H NMR (500 MHz, DMSO-*d*_6_, 298 K): δ (ppm): Dibenzo[*b, f*]oxepine: 10.44 (1H, s, NH), 7.72 (1H, d, *J*_H2,H4_ = 2 Hz, H4), 7.48 (1H, dd, *J*_H1,H2_ = 8.5 Hz, H2), 7.19 (1H, d, H1), 7.09 (1H, d, *J*_H8,H9_ = 8.5 Hz, H9), 6.76 (1H, d, *J*_H6,H8_ = 2.5 Hz, H6), 6.75 (1H, dd, H8), 6.62 (1H, AB spin system, d, *J*_H10,H11_ = 11.5 Hz, H10), 6.57 (1H, AB spin system, d, H11), 3.77 (3H, s, OCH_3_); Azo: 8.19 (2H, d, *J*_H2,H3_ = 8.5 Hz, H3), 7.94 (2H, d, H2), 7.62 (1H, tt, *J*_H3’,H4’_ = 8.5 Hz, *J*_H4’,F_ = 6 Hz, H4’), 7.13 (2H, t, *J*_H3’,F_ = 9 Hz, H3’). 

Ratio *E*/*Z* 4.3:1 

Isomer *E*: ^13^C NMR (125 MHz, DMSO-*d*_6_, 298 K): δ (ppm): 164.55, 164.31, 161.19, 161.16, 157.31, 157.28, 156.44, 156.06, 156.01, 155.95, 154.00, 153.93, 150.91, 148.93, 140.55, 137.55, 134.57, 132.58, 130.20, 130.18, 129.31, 129.24, 129.18, 128.81, 128.57, 128.51, 127.19, 126.09, 125.97, 122.91, 122.90, 122.44, 118.29, 116.87, 116.75, 113.27, 113.24, 113.11, 113.09, 113.01, 112.89, 112.39, 111.22, 111.18, 106.68, 106.66, 55.52, 55.50.

HRMS (ESI): *m*/*z* calculated for C_28_H_19_F_2_N_3_O_3_, 483.13890; found: 483.13899.

**5a.** yield 53%; time of reaction 0.5 h; orange powder mp = 175 °C; 

Isomer *E*: ^1^H NMR (500 MHz, DMSO-*d*_6_, 298 K): δ (ppm): Dibenzo[*b, f*]oxepine: 10.62 (1H, s, NH), 7.79 (1H, d, *J*_H2,H4_ = 2 Hz, H4), 7.67 (1H, dd, *J*_H1,H2_ = 8.5 Hz, H2), 7.28 (1H, d, H1), 7.09 (1H, t, *J*_H8,H7,9_ = 6.5 Hz, H8), 7.08 (1H, dd, *J*_H7,H9_ = 2 Hz, H7), 6.83 (1H, dd, H9), 6.78 (1H, AB spin system, d, *J*_H10,H11_ = 11.5 Hz, H10), 6.72 (1H, AB spin system, d, H11), 3.87 (3H, s, OCH_3_); Azo: 8.47 (1H, t, *J*_H2,H4,6_ = 2 Hz, H2), 8.15 (1H, ddd, *J*_H5,H6_ = 8 Hz, *J*_H4,H6_ = 1 Hz, H6), 8.10 (1H, ddd, *J*_H4,H5_ = 8 Hz, H4), 7.95 (2H, dd, *J*_H2’,H3’_ = 8.5 Hz, *J*_H2’,H4’_ = 1.5 Hz, H2’), 7.77 (1H, t, H5), 7.66–7.61 (3H, m, H3’,H4’).

Isomer *Z*: ^1^H NMR (500 MHz, DMSO-*d*_6_, 298 K): δ (ppm): Dibenzo[*b, f*]oxepine: 10.40 (1H, s, NH), 7.71 (1H, d, *J*_H2,H4_ = 2 Hz, H4), 7.67 (1H, dd, *J*_H1,H2_ = 8.5 Hz, H2), 7.25 (1H, d, H1), 7.09 (1H, t, *J*_H8,H7,9_ = 6.5 Hz, H8), 6.89 (1H, dd, *J*_H7,H9_ = 2 Hz, H7), 6.82 (1H, dd, H9), 6.76 (1H, AB spin system, d, *J*_H10,H11_ = 11.5 Hz, H10), 6.70 (1H, AB spin system, d, H11), 3.86 (3H, s, OCH_3_); Azo: 8.47 (1H, t, *J*_H2,H4,6_ = 2 Hz, H2), 8.15 (1H, ddd, *J*_H5,H6_ = 8 Hz, *J*_H4,H6_ = 1 Hz, H6), 8.10 (1H, ddd, *J*_H4,H5_ = 8 Hz, H4), 7.95 (2H, dd, *J*_H2’,H3’_ = 8.5 Hz, *J*_H2’,H4’_ = 1.5 Hz, H2’), 7.41 (1H, t, H5), 7.66–7.61 (3H, m, H3’,H4’). 

Ratio *E*/*Z* 6:1 

Isomer *E*: ^13^C NMR (125 MHz, DMSO-*d*_6_, 298 K): δ (ppm): 164.79, 164.52, 156.62, 156.60, 153.57, 153.32, 151.85, 151.77, 151.48, 151.45, 144.21, 144.19, 140.84, 140.72, 135.97, 135.29, 131.93, 131.48, 131.46, 130.59, 129.73, 129.69, 129.66, 129.58, 129.25, 128.95, 128.92, 128.77, 127.49, 126.39, 126.09, 126.07, 125.37, 125.09, 122.69, 122.05, 121.82, 120.55, 120.02, 119.93, 116.65, 116.54, 113.27, 113.14, 113.01, 56.16, 56.15.

HRMS (ESI): *m*/*z* calculated for C_28_H_21_N_3_O_3_+ H, 448.16557; found: 448.16562.

**5b**. yield 23%; time of reaction 0.5 h; orange powder mp = 168.5 °C;

Isomer *Z*: ^1^H NMR (500 MHz, DMSO-*d*_6_, 298 K): δ (ppm): Dibenzo[*b, f*]oxepine: 10.62 (1H, s, NH), 7.79 (1H, d, *J*_H2,H4_ = 2 Hz, H4), 7.66 (1H, dd, *J*_H1,H2_ = 8.5 Hz, H2), 7.27 (1H, d, H1), 7.12–7.08 (2H, m, H7, H8), 6.83 (1H, dd, *J*_H8,H9_ = 6 Hz, *J*_H7,H9_ = 3 Hz, H9), 6.78 (1H, AB spin system, d, *J*_H10,H11_ = 11 Hz, H10), 6.72 (1H, AB spin system, d, H11), 3.87 (3H, s, OCH_3_); Azo: 8.46 (1H, t, *J*_H2,H4,6_ = 1.5 Hz, H2), 8.16–8.14 (1H, m, H6), 8.10–8.08 (1H, m, H4), 8.03 (2H, dd, *J*_H2’,H3’_ = 8.5 Hz, *J*_H2’,F_ = 5 Hz, H2’), 7.76 (1H, t, *J*_H5,H4,6_ = 8 Hz, H5), 7.47 (2H, t, *J*_H3’,F_ = 8.5 Hz, H3’).

Isomer *E*: ^1^H NMR (500 MHz, DMSO-*d*_6_, 298 K): δ (ppm): Dibenzo[*b, f*]oxepine: 10.42 (1H, s, NH), 7.71 (1H, d, *J*_H2,H4_ = 2 Hz, H4), 7.66 (1H, dd, *J*_H1,H2_ = 8.5 Hz, H2), 7.25 (1H, d, H1), 6.99–6.96 (2H, m, H7, H8), 6.83 (1H, dd, *J*_H8,H9_ = 6 Hz, *J*_H7,H9_ = 3 Hz, H9), 6.76 (1H, AB spin system, d, *J*_H10,H11_ = 11 Hz, H10), 6.71 (1H, AB spin system, d, H11), 3.86 (3H, s, OCH_3_); Azo: 8.46 (1H, t, *J*_H2,H4,6_ = 1.5 Hz, H2), 8.16–8.14 (1H, m, H6), 7.62–7.59 (1H, m, H4), 8.03 (2H, dd, *J*_H2’,H3’_ = 8.5 Hz, *J*_H2’,F_ = 5 Hz, H2’), 7.43 (1H, t, *J*_H5,H4,6_ = 8 Hz, H5), 7.17 (2H, t, *J*_H3’,F_ = 8.5 Hz, H3’). 

Ratio *E*/*Z* 8.5:1

Isomer *E*: ^13^C NMR (125 MHz, DMSO-*d*_6_, 298 K): δ (ppm): 165.05, 164.77, 164.51, 163.06, 156.62, 156.60, 153.45, 151.65, 151.48, 151.46, 149.50, 149.47, 148.64, 148.61, 144.21, 140.83, 140.71, 135.98, 135.42, 131.48, 131.46, 130.60, 129.73, 129.69, 129.66, 129.25, 129.07, 128.77, 126.47, 126.09, 125.39, 125.15, 125.09, 125.07, 122.76, 122.69, 121.80, 120.55, 119.92, 116.68, 116.65, 116.55, 116.49, 115.95, 115.77, 113.26, 113.16, 113.01, 56.16, 56.15.

HRMS (ESI): *m*/*z* calculated for C_28_H_20_N_3_O_3_+ H, 466.15615; found: 466.15591.

**5c.** yield 87%; time of reaction 0.5 h; orange powder mp =190.5 °C; 

Isomer E: ^1^H NMR (500 MHz, DMSO-*d*_6_, 298 K): δ (ppm): Dibenzo[*b, f*]oxepine: 10.63 (1H, s, NH), 7.78 (1H, d, *J*_H2,H4_ = 2 Hz, H4), 7.66 (1H, dd, *J*_H1,H2_ = 8.5 Hz, H2), 7.27 (1H, d, H1), 7.10–7.09 (2H, m, H7,H8), 6.83 (1H, dd, *J*_H8,H9_ = 6.5 Hz, *J*_H7,H9_ = 3 Hz, H9), 6.77 (1H, AB spin system, d, *J*_H10,H11_ = 11.5 Hz, H10), 6.72 (1H, AB spin system, d, H11), 3.87 (3H, s, OCH_3_); Azo: 8.46 (1H, t, *J*_H2,H4,6_ = 2 Hz, H2), 8.17 (1H, ddd, *J*_H5,H6_ = 8 Hz, *J*_H4,H6_ = 1 Hz, H6), 8.09 (1H, ddd, *J*_H4,H5_ = 8 Hz, H4), 7.88 (1H, td, *J*_H5’,H6’_ = 8.5 Hz, *J*_H6’,F_ = 6.5 Hz, H6’), 7.77 (1H, t, H5), 7.62 (1H, ddd, *J*_H3’,F_ = 9 Hz, *J*_H3’,H5’_ = 2.5 Hz, H3’), 7.31–7.27 (1H, m, H5’).

Isomer *Z*: ^1^H NMR (500 MHz, DMSO-*d*_6_, 298 K): δ (ppm): Dibenzo[*b, f*]oxepine: 10.45 (1H, s, NH), 7.71 (1H, d, *J*_H2,H4_ = 2 Hz, H4), 7.66 (1H, dd, *J*_H1,H2_ = 8.5 Hz, H2), 7.25 (1H, d, H1), 7.01–6.98 (2H, m, H7,H8), 6.82 (1H, dd, *J*_H8,H9_ = 6.5 Hz, *J*_H7,H9_ = 3 Hz, H9), 6.76 (1H, AB spin system, d, *J*_H10,H11_ = 11.5 Hz, H10), 6.71 (1H, AB spin system, d, H11), 3.86 (3H, s, OCH_3_); Azo: 8.46 (1H, t, *J*_H2,H4,6_ = 2 Hz, H2), 8.17 (1H, ddd, *J*_H5,H6_ = 8 Hz, *J*_H4, H6_ = 1 Hz, H6), 8.09 (1H, ddd, *J*_H4,H5_ = 8 Hz, H4), 7.88 (1H, td, *J*_H5’,H6’_ = 8.5 Hz, *J*_H6’,F_ = 6.5 Hz, H6’), 7.48 (1H, t, H5), 7.62 (1H, ddd, *J*_H3’,F_ = 9 Hz, *J*_H3’,H5’_ = 2.5 Hz, H3’), 7.31–7.27 (1H, m, H5’). 

Ratio *E*/*Z* 11:1. 

Isomer *E*: ^13^C NMR (125 MHz, DMSO-*d*_6_, 298 K): δ (ppm): 165.46, 164.68, 164.29, 163.45, 161.01, 158.95, 156.61, 156.59, 153.68, 151.84, 151.47, 151.45, 144.20, 144.18, 140.80, 140.66, 136.91, 136.04, 135.37, 131.48, 131.46, 131.01, 129.81, 129.68, 129.65, 129.25, 129.12, 128.78, 127.34, 126.12, 125.30, 125.09, 123.36, 122.27, 121.52, 120.54, 119.38, 116.65, 116.58, 113.27, 113.19, 113.00, 112.63, 112.10, 105.96, 105.75, 105.56, 105.19, 105.00, 104.79, 56.15.

HRMS (ESI): *m*/*z* calculated for C_28_H_19_F_2_N_3_O_3_+ H, 484.14672; found: 484.14652.

**5d**. yield 36 %; time of reaction 0.5 h; orange powder mp = 179.5 °C. 

Isomer *Z*: ^1^H NMR (500 MHz, DMSO-*d*_6_, 298 K): δ (ppm): Dibenzo[*b, f*]oxepine: 10.64 (1H, s, NH), 7.78 (1H, d, *J*_H2,H4_ = 2 Hz, H4), 7.66 (1H, dd, *J*_H1,H2_ = 8.5 Hz, H2), 7.27 (1H, d, H1), 7.10–7.08 (2H, m, H7,H8), 6.83 (1H, dd, *J*_H8,H9_ = 6.5 Hz, *J*_H7,H9_ = 3 Hz, H9), 6.78 (1H, AB spin system, d, *J*_H10,H11_ = 11.5 Hz, H10), 6.72 (1H, AB spin system, d, H11), 3.87 (3H, s, OCH_3_); Azo: 8.46 (1H, t, *J*_H2,H4,6_ = 2 Hz, H2), 8.21 (1H, ddd, *J*_H5,H6_ = 8 Hz, *J*_H4, H6_ = 1.5 Hz, *J*_H2,H6_ = 1 Hz, H6), 8.07 (1H, ddd, *J*_H4,H5_ = 8 Hz, *J*_H2,H4_ = 1 Hz, H4), 7.79 (1H, t, H5), 7.61–7.59 (1H, m, H4’), 7.37 (2H, t, *J*_H3’,H4’_ = 9 Hz, *J*_H3’,F_ = 9 Hz, H3’). 

Isomer *E*: ^1^H NMR (500 MHz, DMSO-*d*_6_, 298 K): δ (ppm): Dibenzo[*b, f*]oxepine: 10.48 (1H, s, NH), 7.70 (1H, d, *J*_H2,H4_ = 2 Hz, H4), 7.66 (1H, dd, *J*_H1,H2_ = 8.5 Hz, H2), 7.25 (1H, d, H1), 7.10–7.08 (2H, m, H7,H8), 6.82 (1H, dd, *J*_H8,H9_ = 6.5 Hz, *J*_H7,H9_ = 3 Hz, H9), 6.76 (1H, AB spin system, d, *J*_H10,H11_ = 11.5 Hz, H10), 6.71 (1H, AB spin system, d, H11), 3.86 (3H, s, OCH_3_); Azo: 8.46 (1H, t, *J*_H2,H4,6_ = 2 Hz, H2), 8.21 (1H, ddd, *J*_H5,H6_ = 8 Hz, *J*_H4, H6_ = 1.5 Hz, *J*_H2,H6_ = 1 Hz, H6), 7.91 (1H, ddd, *J*_H4,H5_ = 8 Hz, *J*_H2,H4_ = 1 Hz, H4), 7.53 (1H, t, H5), 7.35–7.30 (1H, m, H4’), 7.12 (2H, t, *J*_H3’,H4’_ = 9 Hz, *J*_H3’,F_ = 9 Hz, H3’). 

Ratio *E*/*Z* 1.7:1

Isomer *E*: ^13^C NMR (125 MHz, DMSO-*d*_6_, 298 K): δ (ppm): 164.56, 164.06, 156.58, 155.94, 154.17, 153.89, 152.34, 151.48, 151.45, 150.97, 149.00, 146.02, 144.21, 144.19, 140.78, 140.70, 140.60, 136.06, 135.38, 132.34, 131.61, 131.45, 130.26, 130.19, 130.11, 129.91, 129.68, 129.65, 129.35, 129.26, 128.83, 128.80, 128.19, 126.17, 126.14, 125.10, 124.96, 122.14, 121.02, 120.55, 118.36, 116.68, 116.61, 113.30, 113.24, 113.21, 113.01, 112.51, 56.16.

HRMS (ESI): *m*/*z* calculated for C_28_H_19_F_2_N_3_O_3_+ H, 484.14672; found: 484.14667.

**5e**. yield 80%; time of reaction 0.5 h; orange powder mp = 193.5 °C; 

Isomer *E*: ^1^H NMR (500 MHz, DMSO-*d*_6_, 298 K): δ (ppm): Dibenzo[*b, f*]oxepine: 10.61 (1H, s, NH), 7.80 (1H, d, *J*_H2,H4_ = 2 Hz, H4), 7.57 (1H, dd, *J*_H1,H2_ = 8.5 Hz, H2), 7.23 (1H, d, H1), 7.18 (1H, d, *J*_H8,H9_ = 8.5 Hz, H9), 6.80 (1H, d, *J*_H6,H8_ = 2.5 Hz, H6), 6.77 (1H, dd, H8), 6.64 (1H, AB spin system, d, *J*_H10,H11_ = 11.5 Hz, H10), 6.59 (1H, AB spin system, d, H11), 3.78 (3H, s, OCH_3_); Azo: 8.47 (1H, t, *J*_H2,H4,6_ = 2 Hz, H2), 8.15 (1H, ddd, *J*_H5,H6_ = 8 Hz, *J*_H4,H6_ = 1 Hz, H6), 8.10 (1H, ddd, *J*_H4,H5_ = 8 Hz, H4), 7.95 (2H, dd, *J*_H2’,H3’_ = 8.5 Hz, *J*_H2’,H4’_ = 1.5 Hz, H2’), 7.77 (1H, t, H5), 7.65–7.60 (3H, m, H3’, H4’).

Isomer Z: ^1^H NMR (500 MHz, DMSO-*d*_6_, 298 K): δ (ppm): Dibenzo[*b, f*]oxepine: 10.38 (1H, s, NH), 7.73 (1H, d, *J*_H2,H4_ = 2 Hz, H4), 7.49 (1H, dd, *J*_H1,H2_ = 8.5 Hz, H2), 7.20 (1H, d, H1), 7.17 (1H, d, *J*_H8,H9_ = 8.5 Hz, H9), 6.78 (1H, d, *J*_H6,H8_ = 2.5 Hz, H6), 6.76 (1H, dd, H8), 6.63 (1H, AB spin system, d, *J*_H10,H11_ = 11.5 Hz, H10), 6.57 (1H, AB spin system, d, H11), 3.77 (3H, s, OCH_3_); Azo: 8.47 (1H, t, *J*_H2,H4,6_ = 2 Hz, H2), 8.15 (1H, ddd, *J*_H5,H6_ = 8 Hz, *J*_H4,H6_ = 1 Hz, H6), 7.74 (1H, ddd, *J*_H4,H5_ = 8 Hz, H4), 7.95 (2H, dd, *J*_H2’,H3’_ = 8.5 Hz, *J*_H2’,H4’_ = 1.5 Hz, H2’), 7.42 (1H, t, H5), 7.33–7.29 (3H, m, H3’, H4’).

Ratio *E*/*Z* 5:1 

Isomer *E*: ^13^C NMR (125 MHz, DMSO-*d*_6_, 298 K): δ (ppm): 164.72, 164.44, 161.18, 157.31, 157.29, 156.06, 156.04, 153.59, 153.32, 151.84, 151.79, 140.62, 140.49, 135.94, 135.24, 131.94, 130.55, 130.20, 129.77, 129.58, 129.30, 129.28, 129.00, 128.93, 128.54, 127.50, 127.21, 127.17, 126.32, 126.02, 125.37, 122.93, 122.90, 122.69, 122.18, 121.77, 120.00, 119.78, 116.85, 116.72, 113.00, 112.86, 111.22, 106.68, 106.66, 55.52.

HRMS (ESI): *m*/*z* calculated for C_28_H_21_N_3_O_3_, 447.15774; found: 447,15789.

**5f.** yield 62%; time of reaction 0.5 h; orange powder mp = 188.5 °C; 

Isomer *E*: ^1^H NMR (500 MHz, DMSO-*d*_6_, 298 K): δ (ppm): Dibenzo[*b, f*]oxepine: 10.61 (1H, s, NH), 7.79 (1H, d, *J*_H2,H4_ = 2 Hz, H4), 7.57 (1H, dd, *J*_H1,H2_ = 8 Hz, H2), 7.23 (1H, d, H1), 7.18 (1H, d, *J*_H8,H9_ = 8.5 Hz, H9), 6.80 (1H, d, *J*_H6,H8_ = 2.5 Hz, H6), 6.77 (1H, dd, H8), 6.64 (1H, AB spin system, d, *J*_H10,H11_ = 11.5 Hz, H10), 6.59 (1H, AB spin system, d, H11), 3.78 (3H, s, OCH3); Azo: 8.46 (1H, t, *J*_H2,H4,6_ = 1.5 Hz, H2), 8.14 (1H, ddd, *J*_H5,H6_ = 8 Hz, *J*_H4,H6_ = 1 Hz, H6), 8.09 (1H, ddd, *J*_H4,H5_ = 8 Hz, H4), 8.03 (2H, dd, *J*_H2’,H3’_ = 9 Hz, *J*_H3,F_ = 5 Hz, H2’), 7.77 (1H, t, H5), 7.47 (2H, t, *J*_H3’,F_ = 9 Hz, H3’).

Isomer *Z*: ^1^H NMR (500 MHz, DMSO-*d*_6_, 298 K): δ (ppm): Dibenzo[*b, f*]oxepine: 10.40 (1H, s, NH), 7.73 (1H, d, *J*_H2,H4_ = 2 Hz, H4), 7.50 (1H, dd, *J*_H1,H2_ = 8 Hz, H2), 7.21 (1H, d, H1), 7.17 (1H, d, *J*_H8,H9_ = 8.5 Hz, H9), 6.78 (1H, d, *J*_H6,H8_ = 2.5 Hz, H6), 6.77 (1H, dd, H8), 6.63 (1H, AB spin system, d, *J*_H10,H11_ = 11.5 Hz, H10), 6.57 (1H, AB spin system, d, H11), 3.77 (3H, s, OCH_3_); Azo: 8.46 (1H, t, *J*_H2,H4,6_ = 1.5 Hz, H2), 8.14 (1H, ddd, *J*_H5,H6_ = 8 Hz, *J*_H4,H6_ = 1 Hz, H6), 8.09 (1H, ddd, *J*_H4,H5_ = 8 Hz, H4), 8.03 (2H, dd, *J*_H2’,H3’_ = 9 Hz, *J*_H3,F_ = 5 Hz, H2’), 7.44 (1H, t, H5), 7.18 (2H, t, *J*_H3’,F_ = 9 Hz, H3’). 

Ratio *E*/*Z* 8:1 

Isomer *E*: ^13^C NMR (125 MHz, DMSO-*d*_6_, 298 K): δ (ppm): 165.05, 164.70, 164.42, 163.05, 161.64, 161.18, 159.68, 157.31, 157.29, 156.06, 156.04, 153.46, 151.66, 149.49, 149.47, 148.63, 148.60, 140.61, 140.48, 135.95, 135.37, 130.55, 130.20, 129.77, 129.29, 129.12, 128.54, 127.20, 127.17, 126.40, 126.02, 126.00, 125.38, 125.14, 125.07, 122.92, 122.90, 122.74, 122.67, 121.96, 121.76, 119.76, 116.84, 116.73, 116.67, 116.49, 115.95, 115.77, 112.99, 112.88, 111.21, 106.68, 106.66, 55.51, 54.89.

HRMS (ESI): *m*/*z* calculated for C_28_H_20_FN_3_O_3_, 465.14832; found: 465.14853.

**5g**. yield 89%; time of reaction 0.5 h; orange powder mp = 169.5 °C; 

Isomer *E*: ^1^H NMR (500 MHz, DMSO-*d*_6_, 298 K): δ (ppm): Dibenzo[*b, f*]oxepine: 10.61 (1H, s, NH), 7.79 (1H, d, *J*_H2,H4_ = 2 Hz, H4), 7.56 (1H, dd, *J*_H1,H2_ = 8.5 Hz, H2), 7.23 (1H, d, H1), 7.18 (1H, d, *J*_H8,H9_ = 8.5 Hz, H9), 6.80 (1H, d, *J*_H6,H8_ = 2.5 Hz, H6), 6.77 (1H, dd, H8), 6.64 (1H, AB spin system, d, *J*_H10,H11_ = 11.5 Hz, H10), 6.59 (1H, AB spin system, d, H11), 3.78 (3H, s, OCH_3_); Azo: 8.46 (1H, t, *J*_H2,H4,6_ = 2 Hz, H2), 8.17 (1H, ddd, *J*_H5,H6_ = 8 Hz, *J*_H4,H6_ = 1 Hz, H6), 8.09 (1H, ddd, *J*_H4,H5_ = 8 Hz, H4), 7.87 (1H, td, *J*_H5’,H6’_ = 8.5 Hz, *J*_H6’,F_ = 6 Hz, H6’), 7.78 (1H, t, H5), 7.62 (1H, ddd, *J*_H3’,F_ = 9 Hz, *J*_H3’,F_ = 2.5 Hz, H3’), 7.31–7.27 (1H, m, H5’).

Isomer *Z*: ^1^H NMR (500 MHz, DMSO-*d*_6_, 298 K): δ (ppm): Dibenzo[*b, f*]oxepine: 10.43 (1H, s, NH), 7.73 (1H, d, *J*_H2,H4_ = 2 Hz, H4), 7.49 (1H, dd, *J*_H1,H2_ = 8.5 Hz, H2), 7.20 (1H, d, H1), 7.17 (1H, d, *J*_H8,H9_ = 8.5 Hz, H9), 6.79 (1H, d, *J*_H6,H8_ = 2.5 Hz, H6), 6.76 (1H, dd, H8), 6.63 (1H, AB spin system, d, *J*_H10,H11_ = 11.5 Hz, H10), 6.57 (1H, AB spin system, d, H11), 3.77 (3H, s, OCH_3_); Azo: 8.46 (1H, t, *J*_H2,H4,6_ = 2 Hz, H2), 8.17 (1H, ddd, *J*_H5,H6_ = 8 Hz, *J*_H4,H6_ = 1 Hz, H6), 7.84 (1H, ddd, *J*_H4,H5_ = 8 Hz, H4), 7.87 (1H, td, *J*_H5’,H6’_ = 8.5 Hz, *J*_H6’,F_ = 6 Hz, H6’), 7.49 (1H, t, H5), 7.62 (1H, ddd, *J*_H3’,F_ = 9 Hz, *J*_H3’,F_ = 2.5 Hz, H3’), 7.14–7.07 (1H, m, H5’). 

Ratio *E*/*Z* 5.3:1 

Isomer *E*: ^13^C NMR (125 MHz, DMSO-*d*_6_, 298 K): δ (ppm): 165.37, 164.62, 164.21, 163.45, 161.18, 161.01, 158.95, 157.31, 157.29, 156.06, 156.04, 153.71, 151.86, 140.57, 140.43, 136.92, 136.02, 135.33, 130.95, 130.20, 129.85, 129.30, 129.28, 129.17, 128.55, 127.26, 127.19, 127.16, 126.05, 125.31, 123.38, 122.92, 122.89, 122.22, 121.66, 119.37, 119.29, 116.86, 116.77, 113.01, 112.92, 112.62, 112.11, 111.22, 106.68, 105.96, 105.77, 105.56, 105.19, 105.00, 104.79, 55.51.

HRMS (ESI): *m*/*z* calculated for C_28_H_19_F_2_N_3_O_3_, 483.13890; found: 483,13893.

**5h.** yield 91%; time of reaction 0.5 h; orange powder mp = 167.5 °C; 

Isomer *E*: ^1^H NMR (500 MHz, DMSO-*d*_6_, 298 K): δ (ppm): Dibenzo[*b, f*]oxepine: 10.63 (1H, s, NH), 7.79 (1H, d, *J*_H2,H4_ = 2 Hz, H4), 7.57 (1H, dd, *J*_H1,H2_ = 8.5 Hz, H2), 7.23 (1H, d, H1), 7.18 (1H, d, *J*_H8,H9_ = 8.5 Hz, H9), 6.80 (1H, d, *J*_H6,H8_ = 2.5 Hz, H6), 6.76 (1H, dd, H8), 6.64 (1H, AB spin system, d, *J*_H10,H11_ = 11.5 Hz, H10), 6.59 (1H, AB spin system, d, H11), 3.78 (3H, s, OCH_3_); Azo: 8.46 (1H, t, *J*_H2,H4,6_ = 1.5 Hz, H2), 8.20 (1H, ddd, *J*_H5,H6_ = 8 Hz, *J*_H4,H6_ = 1 Hz, H6), 8.07 (1H, ddd, *J*_H4,H5_ = 8 Hz, H4), 7.79 (1H, t, H5), 7.61 (1H, tt, *J*_H3’,H4’_ = 8.5 Hz, *J*_H4’,F_ = 6 Hz, H4’), 7.37 (2H, t, *J*_H3’,F_ = 9 Hz, H3’).

Isomer *Z*: ^1^H NMR (500 MHz, DMSO-*d*_6_, 298 K): δ (ppm): Dibenzo[*b, f*]oxepine: 10.46 (1H, s, NH), 7.73 (1H, d, *J*_H2,H4_ = 2 Hz, H4), 7.48 (1H, dd, *J*_H1,H2_ = 8.5 Hz, H2), 7.20 (1H, d, H1), 7.17 (1H, d, *J*_H8,H9_ = 8.5 Hz, H9), 6.79 (1H, d, *J*_H6,H8_ = 2.5 Hz, H6), 6.75 (1H, dd, H8), 6.63 (1H, AB spin system, d, *J*_H10,H11_ = 11.5 Hz, H10), 6.57 (1H, AB spin system, d, H11), 3.77 (3H, s, OCH_3_); Azo: 8.46 (1H, t, *J*_H2,H4,6_ = 1.5 Hz, H2), 8.20 (1H, ddd, *J*_H5,H6_ = 8 Hz, *J*_H4,H6_ = 1 Hz, H6), 7.91 (1H, ddd, *J*_H4,H5_ = 8 Hz, H4), 7.54 (1H, t, H5), 7.33 (1H, tt, *J*_H3’,H4’_ = 8.5 Hz, *J*_H4’,F_ = 6 Hz, H4’), 7.12 (2H, t, *J*_H3’,F_ = 9 Hz, H3’).

Ratio *E*/*Z* 3.2:1 

Isomer *E*: ^13^C NMR (125 MHz, DMSO-*d*_6_, 298 K): δ (ppm): 164.49, 163.98, 161.18, 157.31, 157.29, 156.06, 156.03, 154.21, 153.85, 152.35, 150.91, 148.98, 140.56, 140.37, 136.03, 135.33, 132.35, 131.55, 130.19, 129.94, 129.38, 129.29, 129.28, 128.59, 128.56, 128.09, 127.19, 127.15, 126.10, 126.07, 124.96, 122.92, 122.89, 122.09, 121.13, 118.21, 116.88, 116.80, 113.23, 113.04, 112.94, 112.50, 111.23, 106.68, 55.52.

HRMS (ESI): *m*/*z* calculated for C_28_H_19_F_2_N_3_O_3_+ H, 484.14672; found: 484.14640.

As can be seen from the data in Table 1 (see also Appendix A), the smallest energy differences between the ***E*** and ***Z*** isomers occur in the case of compounds with fluorine atoms in positions *ortho* to the azo bond. To check if the obtained compounds show the separation of the n→π* absorption bands, the *UV-VIS* spectra for all products were measured.

To measure the *UV-VIS* spectra of compounds (**4a–h**) and (**5a–h**) solutions of these substances in DMSO with concentrations of 50 and 500 μM were used. Exemplary spectra for *para* and *meta* derivative: (**4d**), (**5d**), and (**5h**) are presented in Figure 3, Figure 4, Figure 5 and Figure 6. Figure 3a shows the *UV-VIS* spectrum of compound (**4d**) with the azo bond in the *para*-position to the amide bond, at a concentration of 50 μM. It shows the characteristic absorption band π→π* responsible, among others, for electron transitions in aromatic systems.

On the other hand, the spectra of substances (**4d**) with concentrations of 500 μM are presented in Figure 3b to observe the area of the expected n→π* band (in the case of magnifying the spectrum for the concentration of 50 μM, the bands are irregular, and the effect is difficult to see). No separation of this absorption band from the π→π* band was observed.

This means that it is not possible to selectively illuminate at a certain frequency and to analyze each of the geometric isomers. In conclusion, no separation of the n→π* absorption band from the π→π* band was observed for all (**4a–h**) compounds.

For compounds (**5**), *UV-VIS* spectra were also recorded at two concentrations of 50 and 500 µM. The spectrum for the lowest concentration of (**5d**) is shown in Figure 4, and the band π→π* can be observed. Figure 5 also presents the *UV-VIS* spectra for substances (**5d**) with an azo bond in the *meta* position with respect to the amide bond with a concentration of 500 μM in the area of the expected n→π* band. It turns out that these absorption bands are separated from the π→π* band, and there is a separation for the ***E*** and ***Z*** isomers, and the difference is 20 nm for a compound with a concentration of 500 μM (when the plot is enlarged for substance (**5d**) with a concentration of 50 μM, the bands are irregular, and it is too low a concentration to observe the effect). In addition, these bands are present in the visible region (violet light). These factors are crucial for the use of these compounds in photopharmacology. A similar but slightly weaker effect (10 nm) can be observed for the compound (**5h**) with the methoxy group in the position *meta* to the oxygen bridge (Figure 6). For products (**5d**) and (**5h**) only, the separation of the n→π* absorption band from the π→π* band was observed. We have concluded that the presence of fluorine in the 2,6 positions not only affects the separation of the π→π* and n→π* bands but also the geometry of the entire system.

## 3. Experimental Section

The general procedure of synthesis of compounds (**4a–h**/**5a–h**):

In a round bottom flask with a capacity of 5 mL was placed 0.2 mmol of the appropriate fluoroazobenzene (**4a–h/5a–h**) and a magnetic stirrer. Then 50 equivalents of thionyl chloride were added. The mixture was heated at 80 °C under a reflux condenser for 30 min. Next, the reaction mixture was evaporated to dryness on a rotary evaporator. 2 mL DCM was added and evaporated again to eliminate residual SOCl_2_. The operation was repeated twice. A total of 0.85 equivalents of methoxydibenzo [b, f] oxepine (**3a** or **3b**) were placed in a vial with a magnetic stirring bar and dissolved in 1 mL of ethyl acetate. The contents were mixed. Then 0.25 mmol of triethylamine was added to the solution. The dry residue from thionyl chloride evaporation was dissolved in 1 ml of ethyl acetate and transferred to a mixing vial. The system was capped and allowed to stir contents overnight. At this time, the reaction mixture was analyzed by the spot position on a TLC plate in hexane-ethyl acetate 7:3 by volume. The mixture was evaporated, dissolved in a little DCM, and then purified by column chromatography in the above system (for more experimental data, see Appendix A).

### 3.1. Nuclear Magnetic Resonance (NMR) Spectroscopy

All the spectra were recorded using a Varian VNMRS spectrometer operating at 11.7 T and Varian Mercury VX 9.4 T magnetic field. Measurements were performed for ca. 1.0 M solutions of all the compounds in DMSO-*d*_6_ or CDCl_3_. The residual signals of DMSO-*d*_6_ (2.54 ppm) and CDCl_3_ (7.26 ppm) in 1H NMR and of the DMSO-*d*_6_ signal (40.45 ppm) and of CDCl_3_ (77.0 ppm) in 13C NMR spectra were used as the chemical shift references. Spin multiplicities are described as s (singlet), d (doublet), t (triplet), q (quartet), m (multiplet), dd (double doublet). Coupling constants are reported in Hertz. All the proton spectra were recorded using the standard spectrometer software and parameters set: acquisition time 3 s, pulse angle 30°. The standard measurement parameter set for 13C NMR spectra was: pulse width 7 s (the 90° pulse width was 12.5 s), acquisition time 1 s, spectral width 200 ppm, 1000 scans of 32 K data point were accumulated and after zero-filling to 64 K; and the FID signals were subjected to Fourier transformation after applying a 1 Hz line broadening. The 1H-13Cgs-HSQC and 1H-13Cgs-HMBC spectra were also recorded using the standard Varian software.

Measurement of NOE effect - 13C NMR spectra with 1H WALTZ decoupling. The use during acquisition decoupling - is the continuous wave (cw) irradiation of one of the proton ortho of dibenzo[*b, f*]oxepine (**4hE**) in the ring closer to the carbonyl group at a single decoupling frequency and observation effect on ^13^C spectrum. And next measurement of the spectrum without irradiation of proton (d_1_ = 8s, at = 1 s) and observation of ^13^C spectrum.

In order to determine the ***E*/*Z*** isomer ratio of 5 h, 1H NMR spectra were recorded after the NMR tube was irradiated with light with a wavelength λ = 525 for 10 min and 60 min.

### 3.2. Mass Spectrometry (MS)

Mass spectra were recorded on spectrometer QTOF Premier firm Waters and spectrometer LTQ Orbitrap Velos.

### 3.3. Photoisomerization Studies by UV-VIS Spectroscopy

EnSpire® multimode plate reader (PerkinElmer, Turku, Finland) with the software EnSpire Workstation version 4.10.3005.1440 (PerkinElmer) in absorbance mode was used for UV-VIS spectroscopic measurements. All experiments were done in at least triplicate. Stock solutions of selected compounds were made in DMSO at appropriate concentrations and stirred at 150 °C overnight to ensure complete relaxation of *E*-isomer. Next 100 μL of stock solution was transferred to 96- a well plate as quickly as possible and absorbance was read.

Afterward, samples were illuminated with light at λ = 525 nm by a self-made LED setup consisting of three high-power 1W LED

(ASMT-AG00-NST00) in series. The working setup is presented in Appendix A. Samples were illuminated with increasing periods until no further spectrum changes were observed.

## 4. Conclusions

The azo compounds with a carboxyl group were transformed into more reactive acid chlorides. These compounds reacted with amine derivatives of dibenzo*[b, f*]oxepine to produce amides, which are dibenzo[*b, f*]oxepine hybrids with an azo bond (**4a–h**) and (**5a–h**). Based on the quantum mechanical calculation, it can be concluded that ***E*** and ***Z*** isomers of these compounds containing fluorine atoms in *ortho*-position to the azo bond are characterized by the lowest values of internal energy, which is associated with an easier transition between two isomeric forms. The overall contribution of the oxepine part to the geometry and the HOMO-LUMO gap vs the substituted azobenzene part seems less significant. In turn, based on the *UV-VIS* spectra, it was established that in the case of compounds with azo bond in *meta*-position, the separation of absorption bands n→π* for both geometric isomers is in the visible part of the spectrum, and their separation from π→π* was also in the visible part of the spectrum. Therefore, these derivatives have the potential to be used in photopharmacology. In order to probably obtain a further shift of the bands towards the red part of the spectrum, it is possible to substitute chlorine atoms into azobenzenes combined with dibenzo [*b, f*] oxepine [20].

## Data Availability

Not applicable.

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
