# Peer review of "Synthesis and Study of Dibenzo[b, f]oxepine Combined with Fluoroazobenzenes—New Photoswitches for Application in Biological Systems"

_molecules, 2022, doi:10.3390/molecules27185836_

Round 1
Reviewer 1 Report (Previous Reviewer 1)
-line 39: "It is a minimally invasive treatment method using visible light and allowing selective destruction of cancer cells" should be reworded, as the mentioned application is only one from the broader context and scope of photopharmacology
-the synthesis of the studied novel hybrid molecules is a simple amide bond formation, the design having a modest level of originality
-the manuscript at places has a descriptive nature, with no strong conclusions being drawn that could be of interest to readers or for the design of improved derivatives. E.g. instead of listing values (lines 116-119), a short presentation and discussion of the results could be more practical. Table 1 and 2 could be better placed in the SI.
-also for Table 3, presenting only illustrative examples in the main text (placing the full data set in the SI) could be enough
-Figure 2 would benefit from some reformatting
-525 nm wavelength used for photoirradiation studies is not sufficiently justified, the result of the photoisomerization would need further clarification (time, E/Z ratio)
-line 332-334: the sentence would need rewording (does it refer to operation at visible wavelength?)
-a conclusion on potential structural modifications to improve photoswitching properties could be of interest to readers
Author Response
Dear Ms. Grace Zhang
Section Managing Editor
Molecules
Manuscript ID: molecules-1870389
Title: Synthesis and study of dibenzo[b, f]oxepine combined with fluoroazobenzenes - new photoswitches for application in biological systems
Authors: Filip Borys, Piotr Tobiasz, Jakub Sobel and Hanna Krawczyk *
Thank you very much for the reviewers’ suggestions for our manuscript. According to your comments, we have posted a revised manuscript.
The answers to Referee’s 1 remarks:
We have marked all of the introduced changes in the text in yellow.
Rev1. Comments and Suggestions for Authors:
-line 39: "It is a minimally invasive treatment method using visible light and allowing selective destruction of cancer cells" should be reworded, as the mentioned application is only one from the broader context and scope of photopharmacology
- Authors’ reply: We have rewritten that sentence in the following manner:
” Photopharmacology covers the projects, synthesis, and application of drugs whose activity can be regulated with light. It is also a minimally invasive treatment method using visible light and allowing selective destruction of cancer cells.”
-the synthesis of the studied novel hybrid molecules is a simple amide bond formation, the design having a modest level of originality
- Authors’ reply: The purpose of this project was to obtain a simple synthesis method of designed compounds new potential photoswitches for application in biological systems.
-the manuscript at places has a descriptive nature, with no strong conclusions being drawn that could be of interest to readers or for the design of improved derivatives. E.g. instead of listing values (lines 116-119), a short presentation and discussion of the results could be more practical. Table 1 and 2 could be better placed in the SI.
- Authors’ reply: In this communication, we present a proposal of new compounds that can be used in photopharmacology. We checked that the introduction of fluoro substituted diphenyldiazene to the dibenzo [b, f] oxepine does not affect on the difference energy between corresponding E / Z isomers and that the total energy for the same substituent in the azo switches is close. The deviation can be observed in pairs (4a / 4e) and (4d / 4h) resulting from the different settings of the methoxy group. We assume that obtained compounds will act on tubulin. We also completed:
“Based on the quantum mechanical calculation it can be concluded that E and Z isomers of obtained compounds, containing fluorine atoms in ortho-position to the azo bond, are characterized by the lowest values of internal energy, which is associated with an easier transition between two isomeric forms.”
We have left Table 1, Table 2 in their original form and in the manuscript text. It shows the size of the energy gap. We did not want to increase the size of already very large SI files.
-also for Table 3, presenting only illustrative examples in the main text (placing the full data set in the SI) could be enough
- Authors’ reply: We have included the full data set of table 3 in SI.
-Figure 2 would benefit from some reformatting
- Authors’ reply: We have reduced figure. 2
-525 nm wavelength used for photoirradiation studies is not sufficiently justified, the result of the photoisomerization would need further clarification (time, E/Z ratio)
- Authors’ reply: The length of 525 nm (harmless visible light) was selected for the tests to check whether E to Z isomerization occurs in this range. Such compounds can be used in photopharmacology as photochromic molecular switches. It is a minimally invasive treatment method using visible light and allowing selective destruction of cancer cells. A very interesting idea was presented by Heck's team [(a) Bleger D.; Hecht, S.,Visible-Light-Activated Molecular Switches. Angew. Chem. Int. Ed. 2015, 54, 11338 – 11349; (b) Bleger, D.; Schwarz, J.; Brouwer, A. M.; Hecht S., o-Fluoroazobenzenes as Readily Synthesized Photoswitches Offering Nearly Quantitative Two-Way Isomerization with Visible Light. J. Am. Chem. Soc. 2012, 134, 20597−20600] and expanded by Feringa and Szymański [(a) Lameijer, L. N.; Budzak, S.; Simeth, N. A.; Hansen, M. J.; Feringa, B. L.; Jacquemin, D.; Szymański, W., General Principles for the Design of Visible-Light-Responsive Photoswitches: Tetra-ortho-Chloro-Azobenzenes. Angew. Chem. 2020, 132, 21847 – 21854; (b) Wegener, M.; Hansen, M. J.; Driessen, A. J. M.; Szymanski, W.; Feringa, B. L., Photocontrol of Antibacterial Activity: Shifting from UV to Red Light Activation. J. Am. Chem. Soc. 2017, 13, 917979−17986; (c) Welleman, I. M.; Hoorens, M. W. H.; Feringa, B. L.; Boersma, H. H.; Szymański, W., Photoresponsive molecular tools for emerging applications of light in medicine. Chem. Sci., 2020, 11, 11672–11691]. The implementation of halogens (fluorine or chlorine) atoms to azo molecules makes it possible to separate the n→π* for stereoisomers E and Z absorption bands in the VIS part of the UV-VIS spectrum and to separate them from the π→π* band, which are in the UV part of the spectrum. This enables selective analysis of each geometric isomer and their selective activation. In this short communication, we present a proposal of new compounds that can be used in photopharmacology. In our further research we will perform further experiments for the obtained compounds, e.g.; photoconversion • composition of the photo-stationary state mixture (PSS),• quantum efficiencies when switching in both directions,• thermal half-life for thermodynamically less stable isomers,• resistance to fatigue,
-line 332-334: the sentence would need rewording (does it refer to operation at visible wavelength?)
- Authors’ reply: We are rewording this sentence: ” In turn, based on the UV-VIS spectra it was established that in the case of compounds with azo bond in meta-position, separation of absorption bands n→π* for both geometric isomers is in visible part of spectrum and their separation from π→π* and was visible in visible part of spectrum.
-a conclusion on potential structural modifications to improve photoswitching properties could be of interest to readers
- Authors’ reply: We have supplemented the following statement in the conclusion:
“In order to probably obtain a further shift of the bands towards the red part of the spectrum, it is possible to substitute chlorine atoms into azobenzenes combined with dibenzo [b, f] oxepine.”
Reviewer 2 Report (Previous Reviewer 2)
Some time ago I already reviewed this manuscript. The authors took into account all the comments and now the article can be published. Of course, thorough proofreading should be done before publication.
Author Response
Dear Ms. Grace Zhang
Section Managing Editor
Molecules
Manuscript ID: molecules-1870389
Title: Synthesis and study of dibenzo[b, f]oxepine combined with fluoroazobenzenes - new photoswitches for application in biological systems
Authors: Filip Borys, Piotr Tobiasz, Jakub Sobel and Hanna Krawczyk *
Thank you very much for the reviewers’ suggestions for our manuscript. According to your comments, we have posted a revised manuscript.
The answers to Referee’s 2 remarks: Thank you very much for the suggestion of reviewers for our manuscript. We have included all the comments of reviewer in our last version no 1822555.

Reviewer 3 Report (New Reviewer)
In this manuscript the authors was synthesized Dibenzo[b,f]oxepine derivatives are an important scaffold in natural, medicinal chemistry and the azo compounds with a carboxyl group were transformed into more reactive acid chlorides. Those compounds reacted with amine derivatives of dibenzo[b,f]oxepine to produce amides Based on the quantum mechanical calculation it can be concluded that E and Z isomers of these compounds, containing fluorine atoms in ortho-position to the azo bond, are characterized by the lowest values of internal energy, which is associated with an easier transition between two isomeric forms. Interestingly they found that the interaction between the obtained isomers of the compounds and the colchicine α and β-tubulin binding site was performed.These results can be summarized in the following previous reviews based on the investigated isomers interact with the 16 colchicine binding site in tubulin with a part of the dibenzo[b, f]oxepine or in a part of the azo switch or both at the same time plays an important role in the in photopharmacology.
I suggest accepting this manuscript for publishing in this molecule journal
Author Response
Dear Ms. Grace Zhang
Section Managing Editor
Molecules
Manuscript ID: molecules-1870389
Title: Synthesis and study of dibenzo[b, f]oxepine combined with fluoroazobenzenes - new photoswitches for application in biological systems
Authors: Filip Borys, Piotr Tobiasz, Jakub Sobel and Hanna Krawczyk *
Thank you very much for the reviewers’ suggestions for our manuscript. According to your comments, we have posted a revised manuscript.
The answers to Referee’s 3 remarks: Thank you very much for the opinion of the reviewer for our manuscript.

Round 2
Reviewer 1 Report (Previous Reviewer 1)
The revision improved the overall quality of the manuscript, however the language is still to be revised and there are still concerns remaining regarding the scope of the presented study vs the journal.
This manuscript is a resubmission of an earlier submission. The following is a list of the peer review reports and author responses from that submission.
Round 1
Reviewer 1 Report
The article aims the synthesis of a small set of dibenzo[b,f]oxepines modified with (fluoro)azobenzenes and investigations thereof. Namely, conformation analysis, HOMO-LUMO level calculation and tubulin interaction were addressed by computational chemistry, UV/VIS absorbance and additionally potential photoisomerization was studied. The authors recently published related studies on azo-dibenzo[b,f]oxepine derivatives (doi: 10.3390/ijms222011033).
The presentation of the results is often not sufficiently clear or lacking details. The scope of the studies and their level of novelty is modest, more suitable to a letter than a full paper. Further experiments and results would be needed to be able to draw meaningful conclusions. The paper in its present form is not likely to attract a wide readership, the overall benefit to publishing this work is not justified. The final conclusion („These derivatives have therefore the potential to be used in photopharmacology” – line 317) is not backed up by the presented experimental results. Further specific comments and concerns are detailed below.
Computational studies: the context of these studies is not discussed in more detail and besides presenting the data, no further meaningful conslusions are drawn. The overall contribution of the oxepine part to the geometry and the HOMO-LUMO gap vs the substituted azobenzene part seems less significant. Fluoro substitued azobenzenes have been studied extensively in the literature, the added value of the present studies is not sufficiently discussed. Regarding the data in Tables 1-2, presenting illustrative examples in the main text and placing the full tables in the Supplementary information could be a better option. Upon discussing the docking results, the substitution pattern is not fully taken into consideration. It is not clear to which compounds the text is referring to in line 178, i.e. which derivatives contain the azo bond in meta position. The docking studies are not verified by pharmacological assays, that could have been a major asset.
Synthesis: a one-step, standard amide formation was used to prepare a small set of novel compounds, in yields ranging from low to medium. A general procedure is presented in the main text for compounds 4a-d/5a-d, however the synthesis of 4e-4h/5e-h is not addressed. The use of the term „approximately” when providing reaction time and volumes is less evident.
NMR: The lack of coalescence at higher temperatures on its own is not a sufficient proof for the presence of E and Z isomers in the mixture (line 218). It is not clear to which structures refers the term ’conformer’ used in lines 209, 217 and 220. Full assignment of the NMR peaks could have been feasible even from mixtures, however this aspect is not addressed. The quantitative ratio of the two set of peaks is not presented (line 207). The Supplementary information states for the NMR spectra „doubled signals from atropisomers”, that is not clear in view of the discussion in the main text (main text lines 214-218) and Figure 1S of the Supplementary information. In general, throughout the discussion it is not evident which structures are discussed (conformers, configurational isomers, etc), that should be improved.
UV: it is not justified why not all compounds were used for UV/VIS measurements and why the spectra were not included in the Supplementary information. Apparently E and Z isomers were not separated and the spectra were measured on E/Z mixtures, however this aspect is not explicitely addressed. Instead of using the spectrum 2 and spectrum 3 reference in the text (line 274,275), referring to the respective schemes would be clearer. The UV/VIS and the photoisomerization results in their present form could be considered more as preliminary results. Line 275: according to Scheme 1, the azo and the amide bonds are not in meta position as stated here. Similarly, it is not clear to which compounds line 315 is referring to („compounds with azo bond in meta-position”), as no such examples are presented in the paper.
The language should be carefully revised, as there are several typos and sentences that would need rewording (e.g. line 11 – oxsepin, line 15 – „Subsequently, modeling…” sentence needs rewording, line 18 – „Based on the…” sentence needs rewording, line 39 – „The very interesting idea…” sentence needs rewording, line 48 – „The connection of…” sentence needs rewording, line 85 – „et all”, line 105 – „For this purpose was using…” sentence needs rewording, line 129 – „In other words…” sentence needs rewording, line 167 – „o”, line 197 – „The goal of the present work…” sentence needs rewording).
The formatting is not consistent, e.g. both letters and numbers are used to indicate references in the text, different fonts are used, etc.
Author Response
Dear Ms. Aiyana Yan
Section Managing Editor
Molecules
Manuscript ID: molecules-1822555
Title: Synthesis and study of dibenzo[b, f]oxepine combined with fluoroazobenzenes - new photoswitches for application in biological systems
Authors: Filip Borys, Piotr Tobiasz, Jakub Sobel and Hanna Krawczyk *
Thank you very much for the suggestion of reviewers for our manuscript. According to your comments, we have posted a revised manuscript.
The answers to Referee’s 1 remarks:
We have marked all introduced changes in the text in yellow.
Rev1. Comments and Suggestions for Authors:
The article aims the synthesis of a small set of dibenzo[b,f]oxepines modified with (fluoro)azobenzenes and investigations thereof. Namely, conformation analysis, HOMO-LUMO level calculation and tubulin interaction were addressed by computational chemistry, UV/VIS absorbance and additionally potential photoisomerization was studied. The authors recently published related studies on azo-dibenzo[b,f]oxepine derivatives (doi: 10.3390/ijms222011033).
The presentation of the results is often not sufficiently clear or lacking details. The scope of the studies and their level of novelty is modest, more suitable to a letter than a full paper. Further experiments and results would be needed to be able to draw meaningful conclusions. The paper in its present form is not likely to attract a wide readership, the overall benefit to publishing this work is not justified. The final conclusion („These derivatives have therefore the potential to be used in photopharmacology” – line 317) is not backed up by the presented experimental results. Further specific comments and concerns are detailed below.
Computational studies: the context of these studies is not discussed in more detail and besides presenting the data, no further meaningful conslusions are drawn. The overall contribution of the oxepine part to the geometry and the HOMO-LUMO gap vs the substituted azobenzene part seems less significant. Fluoro substitued azobenzenes have been studied extensively in the literature, the added value of the present studies is not sufficiently discussed. Regarding the data in Tables 1-2, presenting illustrative examples in the main text and placing the full tables in the Supplementary information could be a better option. Upon discussing the docking results, the substitution pattern is not fully taken into consideration. It is not clear to which compounds the text is referring to in line 178, i.e. which derivatives contain the azo bond in meta position. The docking studies are not verified by pharmacological assays, that could have been a major asset.
- Authors’ reply: In conclusion we added the following sentence:
“The overall contribution of the oxepine part to the geometry and the HOMO-LUMO gap vs the substituted azobenzene part seems less significant.”
We have left Table 2, Table 3 in their original form and in the manuscript text. It shows the size of the energy gap. We did not want to increase the size of already very large SupMat files.
In the discussion about docking results, we presented references (45) about the substitution pattern of colchicine. In table 4 and table 2S we show binding pose and interactions, type of interaction, and active residues correlated with colchicine binding sites. This analysis is an introduction to biological research because we know that the interaction of the obtained compounds with tubulin at the site of the colchicine bond is possible. We of course can report this manuscript as letter.
Synthesis: a one-step, standard amide formation was used to prepare a small set of novel compounds, in yields ranging from low to medium. A general procedure is presented in the main text for compounds 4a-d/5a-d, however the synthesis of 4e-4h/5e-h is not addressed. The use of the term „approximately” when providing reaction time and volumes is less evident.
- Authors’ reply: We thank the reviewer for pinpointing our mistakes. In experimental section, the synthesis applies to all tested compounds: 4a-h and 5a-h. We removed the term "approximately" from this part.
NMR: The lack of coalescence at higher temperatures on its own is not a sufficient proof for the presence of E and Z isomers in the mixture (line 218). It is not clear to which structures refers the term ’conformer’ used in lines 209, 217 and 220. Full assignment of the NMR peaks could have been feasible even from mixtures, however this aspect is not addressed. The quantitative ratio of the two set of peaks is not presented (line 207). The Supplementary information states for the NMR spectra „doubled signals from atropisomers”, that is not clear in view of the discussion in the main text (main text lines 214-218) and Figure 1S of the Supplementary information. In general, throughout the discussion it is not evident which structures are discussed (conformers, configurational isomers, etc), that should be improved.
- Authors’ reply: We have conducted an additional experiment to confirm E-to-Z photoisomerization:
“During the irradiation of (5h) in NMR tube with light at wavelength λ= 525 nm/1h, it obtained one set of signals, the same as in 150oC (E-to-Z photoisomerization). These experiments confirm the presence of E and Z isomers in all products at room temperature (the E isomer prevail in solution). ”
We have fixed all errors related to the terms atropisomer and conformer.
UV: it is not justified why not all compounds were used for UV/VIS measurements and why the spectra were not included in the Supplementary information. Apparently E and Z isomers were not separated and the spectra were measured on E/Z mixtures, however this aspect is not explicitely addressed. Instead of using the spectrum 2 and spectrum 3 reference in the text (line 274,275), referring to the respective schemes would be clearer. The UV/VIS and the photoisomerization results in their present form could be considered more as preliminary results. Line 275: according to Scheme 1, the azo and the amide bonds are not in meta position as stated here. Similarly, it is not clear to which compounds line 315 is referring to („compounds with azo bond in meta-position”), as no such examples are presented in the paper.
- Authors’ reply: We indeed measured the UV-Vis spectra for all compounds. However, in the manuscript, we provided examples for 4d, 5d and 5h. In the case of 4a-4d, 5a-5c, 5e-g no separation of the n → π * band from the π → π * was observed. For compounds 5d and 5h, a separation of these absorption band was observed. It was noticed that these absorption bands are separated from the π → π * band, and there is a separation for the E and Z isomers. We agree with the reviewer that the UV / VIS and the photoisomerization results in their present form could be considered more as preliminary results. We tested compounds with an azo bond in the meta position. We improved the pattern of 5a-h compounds in scheme 1.
The language should be carefully revised, as there are several typos and sentences that would need rewording (e.g. line 11 – oxsepin, line 15 – „Subsequently, modeling…” sentence needs rewording, line 18 – „Based on the…” sentence needs rewording, line 39 – „The very interesting idea…” sentence needs rewording, line 48 – „The connection of…” sentence needs rewording, line 85 – „et all”, line 105 – „For this purpose was using…” sentence needs rewording, line 129 – „In other words…” sentence needs rewording, line 167 – „o”, line 197 – „The goal of the present work…” sentence needs rewording).
The formatting is not consistent, e.g. both letters and numbers are used to indicate references in the text, different fonts are used, etc.
- Authors’ reply: We have improved the text formatting and redesigned the sentences.
Reviewer 2 Report
The article is devoted to the synthesis and computational study of aza-derivatives of Dibenzo[b,f]oxepine. The authors analyzed the energies of the E and Z isomers for all synthesized molecules, gave a brief analysis of the effect of substituents in the phenyl rings and the Dibenzo[b,f]oxepine core on these parameters, in most cases they turned out to be insignificant. The energy difference between fluorine-containing E and Z isomers 5d and 5h turned out to be minimal. Of interest are the results of the docking calculation for these molecules in relation to the colchicine binding site of α and β-tubulin. The effect of substituents on the way the molecule binds to receptors is analyzed. The work contains data that may be useful for further study of the biological properties of these molecules and can be published. However, the work contains a number of shortcomings that need to be corrected. Most of them concern the extremely careless design of the manuscript, which makes it difficult for the reader to read and analyse it.
1) via should be in italics
2) reference numbers must be Arabic numerals and correspond to the format of the journal. The format of the links themselves should also be checked.
3) the separator of the integer and fractional parts must be a dot, in the text it is sometimes a dot, sometimes a comma, and even within one. For example, tables 1 and 3.
4) Scheme 1 with the formulas of the studied compounds should be introduced earlier to make it easier for the reader to analyze the data of the authors. Perhaps it is worth adding columns R1, R2, etc. to the tables to make the influence of the structure clearer.
5) In Table 1, it is necessary to check the number of significant figures in the energy values
6) In the title of Table 1, only "Comparison of total energies of investigation model systems obtained on the B3LYP/ 6-31G* level of theory" should be left. The tiltle is too long.
7) Table 2, perhaps, should be transferred to SupMat, and instead of it, 2 diagrams should be given (for compounds 4 and 5), on which the size of the energy gap should be indicated additionally.
8) The title of table 4 is too long, it is enough to leave "Predicted binding pose of structures 4cZ, 4dE, 4hE, 5bE, 5fE at the colchicine binding site of α and β tubulin", the rest of the information should be given in footnote.
9) In Scheme 1 (p. 7), the reaction conditions should be given.
10) Comments on the effect of substituents on the reaction yield, especially in the case of low yields should be given.
11) On page 9 Scheme 1 and Scheme 2 should be Fig. 3 and fig. 4. Images of spectra should be called "Figure".
12) It is better to add the connection structures to which they refer as an insert to these figures.
13) Figures and tables in the SupMat should have a single numbering (now there are Fig. S1 and Fig. 1S, they are difficult to find).
14) The conclusions need to be rewritten - they reflect what the authors did, but do not tell the main results of their research.
15) The text contains many typos and formatting errors. It should be corrected.
Summary: Authors should improve illustrations and tables in such a way that they reflect the influence of molecular structure on the property under study to the maximum extent possible. They did it best in Table 4. The results should be summarized in Conclusions.
Author Response
Dear Ms. Aiyana Yan
Section Managing Editor
Molecules
Manuscript ID: molecules-1822555
Title: Synthesis and study of dibenzo[b, f]oxepine combined with fluoroazobenzenes - new photoswitches for application in biological systems
Authors: Filip Borys, Piotr Tobiasz, Jakub Sobel and Hanna Krawczyk *
Thank you very much for the suggestion of reviewers for our manuscript. According to your comments, we have posted a revised manuscript.
The answers to Referee’s 2 remarks:
We have marked all introduced changes in the text in blue.
Rev1. Comments and Suggestions for Authors:
The article is devoted to the synthesis and computational study of aza-derivatives of Dibenzo[b,f]oxepine. The authors analyzed the energies of the E and Z isomers for all synthesized molecules, gave a brief analysis of the effect of substituents in the phenyl rings and the Dibenzo[b,f]oxepine core on these parameters, in most cases they turned out to be insignificant. The energy difference between fluorine-containing E and Z isomers 5d and 5h turned out to be minimal. Of interest are the results of the docking calculation for these molecules in relation to the colchicine binding site of α and β-tubulin. The effect of substituents on the way the molecule binds to receptors is analyzed. The work contains data that may be useful for further study of the biological properties of these molecules and can be published. However, the work contains a number of shortcomings that need to be corrected. Most of them concern the extremely careless design of the manuscript, which makes it difficult for the reader to read and analyse it.
- via should be in italics
2) reference numbers must be Arabic numerals and correspond to the format of the journal. The format of the links themselves should also be checked.
3) the separator of the integer and fractional parts must be a dot, in the text it is sometimes a dot, sometimes a comma, and even within one. For example, tables 1 and 3.
- Authors’ reply: We have corrected the above errors.
4) Scheme 1 with the formulas of the studied compounds should be introduced earlier to make it easier for the reader to analyze the data of the authors. Perhaps it is worth adding columns R1, R2, etc. to the tables to make the influence of the structure clearer.
- Authors’ reply: We have introduced Scheme 1 with the formulas earlier to make it easier for the reader to analyze the data. We did not introduce the columns R1, R2 etc. to the tables, so as not to increase the volume of the tables.
5) In Table 1, it is necessary to check the number of significant figures in the energy values
6) In the title of Table 1, only "Comparison of total energies of investigation model systems obtained on the B3LYP/ 6-31G* level of theory" should be left. The title is too long.
8) The title of table 4 is too long, it is enough to leave "Predicted binding pose of structures 4cZ, 4dE, 4hE, 5bE, 5fE at the colchicine binding site of α and β tubulin", the rest of the information should be given in footnote.
- Authors’ reply: We corrected Table 1 and the title of Table 1 and of Table 4.
7) Table 2, perhaps, should be transferred to SupMat, and instead of it, 2 diagrams should be given (for compounds 4 and 5), on which the size of the energy gap should be indicated additionally.
- Authors’ reply: We have left Table 2 in original form and in the manuscript text. It shows the size of the energy gap. We did not want to increase the size of already very large SupMat files.
9) In Scheme 1 (p. 7), the reaction conditions should be given.
- Authors’ reply: We have completed the reaction conditions in scheme 1.
10) Comments on the effect of substituents on the reaction yield, especially in the case of low yields should be given.
- Authors’ reply: We have commented :
“Summarizing the results for the obtained synthesis products, we can observe higher yields for products 5 than 4 (except 4b and 5b). This is probably related to the greater electrophilicity of the acid chloride carbonyl atom in the 2a-2d substrates and the electron density distribution in the ring related to the azo bond position.”
11) On page 9 Scheme 1 and Scheme 2 should be Fig. 3 and fig. 4. Images of spectra should be called "Figure".
- Authors’ reply: We have changed the names for spectra 1,2,3 and 4 to figures 3,4,5 and 6.
12) It is better to add the connection structures to which they refer as an insert to these figures.
- Authors’ reply: We have added the connection structures in figures 3,4 and 6.
13) Figures and tables in the SupMat should have a single numbering (now there are Fig. S1 and Fig. 1S, they are difficult to find).
- Authors’ reply: We have corrected the numbering for figures 1S and fig 2S and tables 1S and 2S in the SupMat.
14) The conclusions need to be rewritten - they reflect what the authors did, but do not tell the main results of their research.
15) The text contains many typos and formatting errors. It should be corrected.
Summary: Authors should improve illustrations and tables in such a way that they reflect the influence of molecular structure on the property under study to the maximum extent possible. They did it best in Table 4. The results should be summarized in Conclusions.
- Authors’ reply: We have rewritten the conclusions according to the reviewer's comments and we have corrected typos and formatting errors.
“It can be observed higher yields for products 5 than 4 (except 4b and 5b). This is probably related to the greater electrophilicity of the acid chloride carbonyl atom in the (2a-2d) substrates and the electron density distribution in the ring related to the azo bond position.”

Reviewer 3 Report
The paper describes the synthesis of a series of molecules targeting photoswitching behavior. The work was conducted as a complete story, yet the synthesis of the target molecules is relatively simple and does not have high novelty. Reconsideration is required after the following questions/suggestions are addressed:
1. Why 525 nm was chosen needs to be explained. It seems most of the compounds do not have absorbancy at 525 nm. Why 525 nm is inducing E to Z photoisomerization and to what extent did the photoisomerization happen need to be proved. This would be the key question.
2. The computational calculation can be improved by using a more accurate basis set. Additional TDDFT calculation for simulated UV-Vis spectra can help to understand the assignment of band assignments.
3. Table 1 includes the total energy for the compounds that do not carry any information. This part should be kept in SI and a table with the first and last column would be enough.
4. Table 2 is positioned in an odd place and cuts the paragraph in half. The formatting needs to be improved. Also, Table 2 can be made into a figure, which can be a lot more clear for readers to compare them to each other.
5. On page 6, line 173, the author claims “free energy”. Free energy requires to compute vibrations. It is not free energy that is being discussed here.
6. A lot of details of computation or experiment, such as “saved in Mol2 format” and a lot more should be kept in SI. The main text is very long already.
7. The numbering system for figures in SI is a bit awkward (Fig. 1S and Fig. S1 coexist).
8. The language needs to be polished. There are a lot of small issues with how the manuscript is written.
Author Response
Dear Ms. Aiyana Yan
Section Managing Editor
Molecules
Manuscript ID: molecules-1822555
Title: Synthesis and study of dibenzo[b, f]oxepine combined with fluoroazobenzenes - new photoswitches for application in biological systems
Authors: Filip Borys, Piotr Tobiasz, Jakub Sobel and Hanna Krawczyk *
Thank you very much for the suggestion of reviewers for our manuscript. According to your comments, we have posted a revised manuscript.
The answers to Referee’s 3 remarks:
We have marked all introduced changes in the text in green .
The paper describes the synthesis of a series of molecules targeting photoswitching behavior. The work was conducted as a complete story, yet the synthesis of the target molecules is relatively simple and does not have high novelty. Reconsideration is required after the following questions/suggestions are addressed:
- Why 525 nm was chosen needs to be explained. It seems most of the compounds do not have absorbancy at 525 nm. Why 525 nm is inducing E to Z photoisomerization and to what extent did the photoisomerization happen need to be proved. This would be the key question.
- Authors’ reply: The length of 525 nm (harmless visible light) was selected for the tests to check whether E to Z isomerization occurs in this range. Such compounds can be used in photopharmacology as photochromic molecular switches. It is a minimally invasive treatment method using visible light and allowing selective destruction of cancer cells. A very interesting idea was presented by Heck's team [(a) Bleger D.; Hecht, S.,Visible-Light-Activated Molecular Switches. Angew. Chem. Int. Ed. 2015, 54, 11338 – 11349; (b) Bleger, D.; Schwarz, J.; Brouwer, A. M.; Hecht S., o-Fluoroazobenzenes as Readily Synthesized Photoswitches Offering Nearly Quantitative Two-Way Isomerization with Visible Light. J. Am. Chem. Soc. 2012, 134, 20597−20600] and expanded by Feringa and Szymański [(a) Lameijer, L. N.; Budzak, S.; Simeth, N. A.; Hansen, M. J.; Feringa, B. L.; Jacquemin, D.; Szymański, W., General Principles for the Design of Visible-Light-Responsive Photoswitches: Tetra-ortho-Chloro-Azobenzenes. Angew. Chem. 2020, 132, 21847 – 21854; (b) Wegener, M.; Hansen, M. J.; Driessen, A. J. M.; Szymanski, W.; Feringa, B. L., Photocontrol of Antibacterial Activity: Shifting from UV to Red Light Activation. J. Am. Chem. Soc. 2017, 13, 917979−17986; (c) Welleman, I. M.; Hoorens, M. W. H.; Feringa, B. L.; Boersma, H. H.; Szymański, W., Photoresponsive molecular tools for emerging applications of light in medicine. Chem. Sci., 2020, 11, 11672–11691]. The implementation of halogens (fluorine or chlorine) atoms to azo molecules makes it possible to separate the n→π* for stereoisomers E and Z absorption bands in the VIS part of the UV-VIS spectrum and to separate them from the π→π* band, which are in the UV part of the spectrum. This enables selective analysis of each geometric isomer and their selective activation.
- The computational calculation can be improved by using a more accurate basis set. Additional TDDFT calculation for simulated UV-Vis spectra can help to understand the assignment of band assignments.
- Authors’ reply: In this manuscript, we analyzed the geometry of the E / Z (4a-4h) and (5a-5h) isomers in azo molecules to show the energetic effect of isomerization. Of course, in our further research, we are planning to perform more detailed TDDFT calculations for the simulated UV-Vis spectra and compare them with the experimental data.
- Table 1 includes the total energy for the compounds that do not carry any information. This part should be kept in SI and a table with the first and last column would be enough.
- Authors’ reply: In Table 1, we decided to present all the data. We did not want to duplicate data and to increase the size of already very large SupMat files.
- Table 2 is positioned in an odd place and cuts the paragraph in half. The formatting needs to be improved. Also, Table 2 can be made into a figure, which can be a lot more clear for readers to compare them to each other.
- Authors’ reply: We have corrected the position of table 2. We have left Table 2 in the original form and in the manuscript text. It shows the size of the energy gap. We do not want to enlarge the already extensive SM.
- On page 6, line 173, the author claims “free energy”. Free energy requires to compute vibrations. It is not free energy that is being discussed here.
- Authors’ reply: We have corrected the sentence as follows:
“It is worth noting that for all the compounds obtained, the binding energy in the complex with tubulin was lower than for the colchicine itself.”
- A lot of details of computation or experiment, such as “saved in Mol2 format” and a lot more should be kept in SI. The main text is very long already.
- Authors’ reply Table 4 presents the predicted binding pose of structures only for 4cZ, 4dE, 4hE, 5bE, 5fE at the colchicine binding site of α and β tubulin. It is only part of the data (all in SM). We have not transfered this data to SM and left it in the main text to make the description easier for the reader.
- The numbering system for figures in SI is a bit awkward (Fig. 1S and Fig. S1 coexist).
- Authors’ reply: We have corrected the numbering for figures 1S and fig 2S in the SupMat.
- The language needs to be polished. There are a lot of small issues with how the manuscript is written.
- Authors’ reply: We have polished language once again.
Round 2
Reviewer 1 Report
Several shortcomings of the presentation of the work and the results were corrected. However, regarding the scope of the work publication in Molecules is still not fully justified.
Specific comments are listed below:
i) Scheme 1 should be checked: in 2a-d there is a COOH in ortho position, whereas in 5a-5h there is a COOH in meta position; on the arrow EtOaAc instead of EtOAc
ii) line 130: "Direct photo-excitation..." wording should be corrected
iii) line 213, line 323: comment on synthetic yields is not necessary, would suggest deleting it
iv) line 228: interpretation of NMR result upon heating should be fine-tuned (i.e. spectrum at 150°C)
v) for NMR spectra it would be very informative to include the E/Z ratio at rt
vi) interpretation of the photochemical results could be improved (i.e. comments on the full set of compounds, structure-property relationship, if relevant)
vii) biological assays (if available) besides the computational studies would be a major asset for the presented work
Reviewer 3 Report
1. The literatures the author provided shows a clear change in UV-VIS and can be characterized to give the component of the spectra. This work however, failed to give a quantitative discription for the degree of the E-Z transition
7. figures 1S and fig 2S is still not the same format. Either Figure or Fig.